# Time-Series Classification Based on Fusion Features of Sequence and Visualization

**Baoquan Wang [1,2,3], Tonghai Jiang [1,3,*], Xi Zhou [1,3], Bo Ma [1,2,3] and Fan Zhao [1,3] and Yi Wang [1,3]**

[1]  The Xinjiang Technical Institute of Physics & Chemistry, CAS, Urumqi 830011, China;
    wangbaoquan14@mails.ucas.ac.cn (B.W.); zhouxi@ms.xjb.ac.cn (X.Z.); mabo@ms.xjb.ac.cn (B.M.);
    zhaofan@ms.xjb.ac.cn (F.Z.); wangyi@ms.xjb.ac.cn (Y.W.)
[2]  University of Chinese Academy of Sciences, Beijing 100049, China
[3]  Xinjiang Laboratory of Minority Speech and Language Information Processing, Urumqi 830011, China
[*]  Correspondence: jth@ms.xjb.ac.cn; Tel.: +86-099-1383-7795

**Abstract:** For the task of time-series data classification (TSC), some methods directly classify raw time-series (TS) data. However, certain sequence features are not evident in the time domain and the human brain can extract visual features based on visualization to classify data. Therefore, some researchers have converted TS data to image data and used image processing methods for TSC. While human perceptionconsists of a combination of human senses from different aspects, existing methods only use sequence features or visualization features. Therefore, this paper proposes a framework for TSC based on fusion features (TSC-FF) of sequence features extracted from raw TS and visualization features extracted from Area Graphs converted from TS. Deep learning methods have been proven to be useful tools for automatically learning features from data; therefore, we use long short-term memory with an attention mechanism (LSTM-A) to learn sequence features and a convolutional neural network with an attention mechanism (CNN-A) for visualization features, in order to imitate the human brain. In addition, we use the simplest visualization method of Area Graph for visualization features extraction, avoiding loss of information and additional computational cost. This article aims to prove that using deep neural networks to learn features from different aspects and fusing them can replace complex, artificially constructed features, as well as remove the bias due to manually designed features, in order to avoid the limitations of domain knowledge. Experiments on several open data sets show that the framework achieves promising results, compared with other methods.

**Keywords:** time series data; classification; fusion feature; visualization; area graph; attention

## 1. Introduction

Time-series (TS) data is a set of values sequentially ordered in time, which is seen frequently in real-life, such as financial data, trajectory data, weather data, and so on. With the development and application of the Internet of Things (IoT), the data collected by various sensors is also TS data. Research on TS data is diverse, such as compression, storage and query, anomaly detection, prediction, and so on [1]. This paper focuses on the time-series classification (TSC) task, the purpose of which is to classify concrete TS data into pre-determined categories which have similar characteristics.

For the task of TSC, our predecessors have done a lot of research and produced many methods. These methods are mainly based on distance functions, such as dynamic time warping (DTW) [2]; features, such as the shapelet transform (ST) [3]; and ensemble methods, such as the hierarchical vote COTE (HIVE-COTE) [4], combining the former two types of methods.

With the development and application of deep learning, a large number of methods for this task have also been proposed. Researchers typically use various transformation methods or artificially constructed features to input into deep neural networks (DNN) and train a model for TSC; yet, some methods directly use DNN, such as long short-term memory (LSTM) [5], to process raw TS data and learn an end-to-end classification model.

Human beings can use visual observation to distinguish the differences between TS records, according to key features, and classify them. Visualization is also very important in scientific research. In the field of natural language processing (NLP), Chinese character representations have been improved by using the pictographic meanings of Chinese characters [6]. In the field of TS research, features extracted by visualization have been used for the task of prediction [7]. Among deep learning methods for TSC, a type of method converts TS data into image data, then use some image processing methods to classify the TS data. Representative transformation methods include Gramian fields [8,9], recurrence plots (RP) [10,11], Markov transition fields (MTF) [12], and so on.

Usually, a human's perception of things is a combination of human senses from different aspects, while the deep learning methods for TSC proposed so far have either used only sequential features or visualization features. In the field of few-shot learning, scholars have used DNNs to extract data features from different perspectives, in order to improve the performance of deep learning models on small training data sets [13,14].

Enlightened by these advances, we propose a framework, TSC-FF, based on deep learning to learn features from different aspects, which enhances the feature space and replaces complex, artificially constructed features, in order to remove the bias due to manually designed features, as well as to avoid the limitations of domain knowledge. Specifically, in order to enable the network to learn the most discriminant useful features for the classification task, we use well-trained LSTM-A neural networks and CNN-A neural networks to extract the features of TS data. LSTM-A is used to extract sequence features, while CNN-A is used to extract visualization features after the TS data is visualized.

Our approach differs from the existing visual transformation methods, such as Gramian fields [8,9], RP [10,11], and MTF [12]. In order to imitate human vision (i.e., what we see is what we get), we take the most direct visualization method, transforming the TS data into an Area Graph. Area Graphs are a type of Line Graph, but with the area below the line filled in with a certain colour or texture. Compared with the existing visualization methods, Area Graph is simpler and avoids the loss of some information in complex conversions and additional computational cost. In addition, an attention mechanism is used to enhance the features of key points, which is generated from contributing regions in the data for the specific labels.

In this paper, experiments are carried out on the University of California at Riverside (UCR) collection [15]. The experimental results show that the proposed method can achieve satisfactory results. Our research in this paper significantly differs from previous work in the following aspects:

- The framework imitates the mechanism of the human brain, in that the human brain's cognition is a combination of the human body's multiple senses. Thus, we use a DNN to learn features from different aspects to enhance the feature space. Then, based on the fusion of features, we carry out the TSC task and gain promising results.
- The framework uses well-trained LSTM-A and CNN-A to extract sequence features and visualization features, as well as combining an attention mechanism to extract the key features that contribute to classification. In particular, an innovational category trend attention (CT-Attention) is gained from data belonging to the same category in an innovative way.
- The framework transforms TS data into Area Graphs. Compared with the existing visualization methods (such as RP), this conversion method is simpler and avoids the loss of some information in complex conversions as well as additional computational cost.

The rest of this paper is organized as follows: In Section 2, we present the related work. Section 3 describes the proposed method in detail, including the structure of the network and how to calculate the attention to find out the contributing region in the raw data for the specific labels. Section 4 presents

the evaluation results. Finally, we present conclusions, discussions, and suggestions for future research in Section 5.

## 2. Related Work

### 2.1. Related Work on TSC

There are a variety of methods for the task of TSC. These methods can be divided into four main categories: distance-based methods, feature-based methods, ensemble methods, and deep learning methods.

**Distance-based methods.** This type of method uses a variety of distance functions [16] to measure the similarity between TS records for classification. The most widely used distance function methods are DTW and its variants, such as the one nearest neighbor (1NN) classifier with DTW (1NN-DTW) [2]. In [17], the authors proposed a framework named Proximity Forest (PF), which uses Proximity Trees with 11 distance measures for the TSC task. The main drawback of this kind of method is the huge computational cost involved.

**Feature-based methods.** The basis of this type of method is a variety of features learned from TS data, through which we can distinguish the differences between data and classify them. The methods in this class include ST [18,19], bag of symbolic Fourier approximation (SFA) symbols (BOSS) [20], time-series forest (TSF) [21], and TS classification based on a bag-of-features representation (TSBF) [22]. Word ExtrAction for time SEries cLassification (WEASEL) [23] uses a novel discriminative feature generation and a feature selection method based on bag-of-patterns (BOP). The Shapelet Transform Classification (STC) uses a novel way to find shapelet and increases accuracy for multi-class problems [24]. Random Interval Spectral Ensemble (RISE) [25] combines tree structure with multiple features for TSC. Some methods input hand-engineered features using some domain knowledge into a DNN discriminative classifier. The disadvantages of these methods lie in the complexity and weak generality of building features, which obviously limits their versatility. Besides hand-engineered features, some methods use a DNN to extract the features of TS for classification: In [26,27], the authors added a deconvolutional operation into a convolutional neural network (CNN)-based model to reconstructing a multivariate time-series. Deep Belief Networks (DBNs) [28] and Recurrent Neural Network auto-encoders [29] have also been used to model the latent features in an unsupervised manner. Other studies [30,31] have used self-predicting modeling to ensure the effectiveness of feature learning. Inspired by computer vision processing methods, some scholars have converted TS data into image data and used image processing methods to extract features for TSC. Typical image transform methods include Gramian Angular Field (GAF) [8,9], RP [10,11], Markov Transition Fields (MTF) [12], and so on. However, these methods only use a single kind of feature.

**Ensemble methods.** This kind of method combines several effective methods, in order to obtain the most appropriate classification results from the classification results obtained through different mechanisms. It includes three typical methods: Elastic Ensemble (EE) [32], the collection of transformation ensembles (Flat-COTE) [33], and HIVE-COTE [4]. Among them, EE integrates 13 classification methods based on distance measurement; Flat-COTE includes 35 classification methods, where several feature-based classification methods are added in Flat-COTE; while HIVE-COTE is based on Flat-COTE, improving the mechanism of getting the final classification result from each sub-classification result. Time Series Combination of Heterogeneous and Integrated Embedding Forest (TS-CHIEF) [34] rivals HIVE-COTE in accuracy but is faster than HIVE-COTE. Random Convolutional Kernel Transform (ROCKET) [35] claims to achieve optimal performance with less computational cost. At the cost of a slight decrease in accuracy, Canonical Time-series Characteristics (Catch22) [36] generates a diverse and interpretable feature set with a greatly reduced number according to the properties of the TS data. The basis of these methods is still distance-based or feature-based; although the drawbacks of both types of methods are alleviated by ensembling, their flaws still exist. In addition,

as a variety of classification models and features are ensembled, the model is very complex, which limits its practical application.

**Deep learning methods.** A variety of deep learning models have been proposed for TSC. This type of method trains a DNN model to form a mapping from data to categories. These models are discriminative, where they input raw TS data and output a probability distribution over the class variables in a data set. The models include Multi-scale CNN (MVCNN) [37], fully convolutional networks (FCN) [27], and deep residual networks (ResNet) [38], as well as many hybrid models such as attention-based LSTM-CNNs [39], multivariate LSTM-FCNs [40], and LSTM-FCNs [41]. It is worth noting that InceptionTime [42] claims higher accuracy and being faster than HIVE-COTE.

### 2.2. Related Work on Feature Extraction through DNN from Different Aspects

Deep learning has made amazing progress in many fields. However, the impact of changing the structure of the network on classification accuracy has been getting smaller and smaller and, so, researchers have begun to focus on expanding data sets, as the scale of the existing common data sets is inadequate compared to the current level of deep learning development. However, expanding data sets is not a simple task, as manual marking is needed, in general, because artificial participation leads to errors. These errors inevitably become factors that affect the training effect. Another method is to use more effective features. However, artificial features can only achieve good results in specific tasks.

A DNN imitates the mechanisms of the human brain [43] and is able to automatically learn the characteristics of data without heavy pre-processing for further processing. However, a human's understanding of things is often a combination of multiple human senses; therefore, some scholars have used DNN to extract features from different perspectives for some tasks. In [13], the authors trained two VAEs to extract visual and semantic features of images, respectively, for generalized zero- and few-shot learning. In [14], the authors used a pre-trained saliency model to segment the foreground and background, and trained two feature extractors—one for extracting foreground features and the other for extracting background features—to improve image classification. In the field of NLP, the authors used historical Chinese scripts to enrich the pictographic evidence in characters and design CNN structures tailored to Chinese character image processing. The proposed glyph-based models gained outstanding results in multiple Chinese NLP tasks [6]. In the field of TS data mining, TS have been converted to bar images and a CNN was used for TS prediction, the results of which was also promising [7].

### 3. Model Design

The proposed framework consists of two parts, LSTM-A and CNN-A (as shown in Figure 1). LSTM-A is pre-trained with the loss function $AUX_1$ on raw TS data and CNN-A is pre-trained with the loss function $AUX_2$ on Area Graphs converted from the raw TS data. After LSTM-A and CNN-A are well-trained, raw test TS data are input to the LSTM-A to extract sequence features and corresponding Area Graphs to CNN-A for visualization features. Fusion features are obtained through a fully-connected layer concatenating the sequence features and visualization features. The final classification operation is realized by a softmax layer. An Attention mechanism is used to make the model focus on key sub-sequences and sub-regions which contain more discriminative information for TSC.

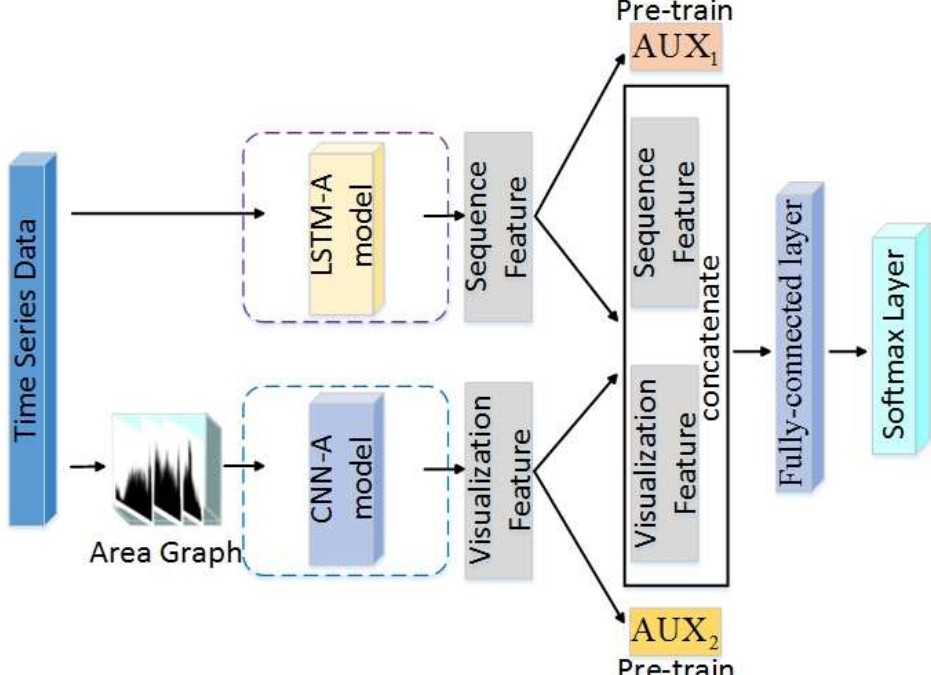

**Figure 1.** Overview of the framework. $AUX_1$ and $AUX_2$ are loss functions in the pre-training stage.

### 3.1. Data Notation

We use $D = \{(x_i, y_i)\}_{i=1}^{N}$ to represent a time-series database, where $N$ represents the number of records, $x_i = \{x^1, x^2 \cdots x^m\}$ represents the $i^{\text{th}}$ record, $x^j$ represents the ordered observation value of the record in the whole time-series $m$, and $y_i \in \{1, 2 \cdots C\}$ represents the category of the $i^{\text{th}}$ record, where $C \in Z^+$ is the number of classes. Generally speaking, the goal of TSC is to learn a mapping function from $x_i$ to $y_i$, as shown in the following formula:

$$y_i = f(x_i). \tag{1}$$

where $f(\cdot)$ is the mapping function we aim to learn. The following two sections describe how to extract sequence features and visualization features through LSTM-A and CNN-A, respectively.

### 3.2. LSTM-A

An RNN is a type of neural network used to process sequence data. There are many variants of RNNs, some representative ones including LSTM, Gated Recurrent Unit (GRU) [44], Bi-directional RNN [45], and so on. Greff et al. [46] compared popular RNN variants, showing that the RNN variants have almost the same performance and that LSTM is superior to other simplified LSTMs (such as GRU). In [47], the authors tested more than 10,000 RNN structures and found that, in certain tasks or situations, some RNN variants work better than LSTM; but only in special cases. Comparing GRU and LSTM, on the one hand, GRU has fewer parameters, so its training is faster and requires less data to generalize. On the other hand, if given enough training data, the great expressive power of LSTM may produce better results than GRU. In addition, in the field of TSC research, many studies have chosen LSTM [39–41] to do TSC and achieved promising results. So, we chose LSTM to learn the temporal dependencies of TS data. However, the temporal dependencies of long input sequences cannot be reasonably learned and, so, we added attention mechanisms to learn these long-term dependencies [48]. Finally, we used the LSTM-A model (Figure 2) to learn sequence features.

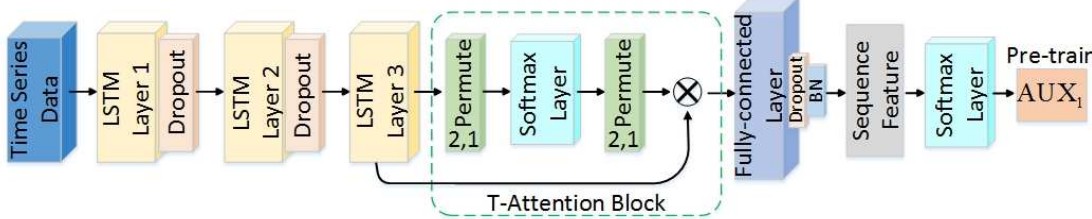

**Figure 2.** Architecture of the LSTM-A model. T-Attention means temporal attention block, the permute layers are used to permute the dimensions of the input matrix, and BN stands for batch normalization.

Given a time-series data set $D$, the LSTM-A model processes the records as follows:

$$z = tanh(\overbrace{x^t, h^{t-1}}),\tag{2a}$$

$$z^f = sigmoid(\overbrace{x^t, h^{t-1}}),\tag{2b}$$

$$z^i = sigmoid(\overbrace{x^t, h^{t-1}}),\tag{2c}$$

$$z^o = sigmoid(\overbrace{x^t, h^{t-1}}),\tag{2d}$$

$$c^t = c^{t-1} \odot z^f \oplus z^i \odot z,\tag{2e}$$

$$h^t = z^o \odot tanh\left(c^t\right).\tag{2f}$$

where $h^{t-1}$ and $c^{t-1}$ are the outputs of the last LSTM cell; $x^t$ is the current input; $\frown$ stands for the splicing operation; $z^i$, $z^f$, and $z^o$ are the input, forget, and output gates obtained by different parameters, respectively; $\odot$ and $\oplus$ represent matrix multiplication and matrix addition, respectively; and $h^t$ and $c^t$ are the outputs of the current LSTM cell.

The T-Attention block is used to enhance the performance of sequence feature learning for very long input records. Its calculation formula is as follows:

$$M^t = tanh\left(w^h h^t\right),\tag{3a}$$

$$a^t = softmax\left(w^t M^t\right),\tag{3b}$$

$$F^s = FC\left(a^t \otimes h^t\right),\tag{3c}$$

$$P^s = softmax\left(F^s\right),\tag{3d}$$

$$AUX_1 = H\left(P^s, Y\right) = -\sum_i y_i log\left(p_i^s\right).\tag{3e}$$

where $h^t$ is the output of the $t^{\text{th}}$ hidden unit of the last LSTM layer, $a^t$ is the $t^{\text{th}}$ attention weight, $w^h$ and $w^t$ are weighted matrices, $a^t$ and $h^t$ are merged, and $\otimes$ represents the matrix merge operation. Then, we use a Fully-connected layer to transform it to obtain sequence features $F^s$. To prevent overfitting and gradients from disappearing, we use dropout and Batch Normalization continuously after the Fully-connected layer. Finally, a softmax layer is used to obtain the classification results, where $P^s$ is the output prediction probability sequence of LSTM-A. In the pre-training phase of LSTM-A, we use cross-entropy as the loss function $AUX_1$, $Y$ is the true probability sequence, and $p_i^s$ and $y_i$ are the predicted probability and true probability of the record belonging to category $i$, respectively.

### 3.3. CNN-A

TS data is a series of sequential data. It is difficult for human beings to classify them only by ordered numbers, as vision is an important sense for human beings. By visualizing TS data, humans can easily classify them. Deep learning is a technology that imitates the human brain and, so, we consider visualizing time-series data and extracting features to imitate human vision for TSC. In addition,

through data visualization, human vision can notice the parts with the largest differences and classify them. This is similar to the attention mechanism in deep learning. Therefore, we added an attention mechanism to CNN to extract key visualization features.

We used CNN-A to imitate human vision (the network structure is shown in Figure 3). For univariate TS data $D$, we convert each TS record $x_i$ into a black-and-white Area Graph (examples shown in Figure 4a,b) as input. In order to avoid the influence of unnecessary information such as co-ordinates, we removed the co-ordinates and other information from the Area Graphs, and only retained the graph part showing the data fluctuations.

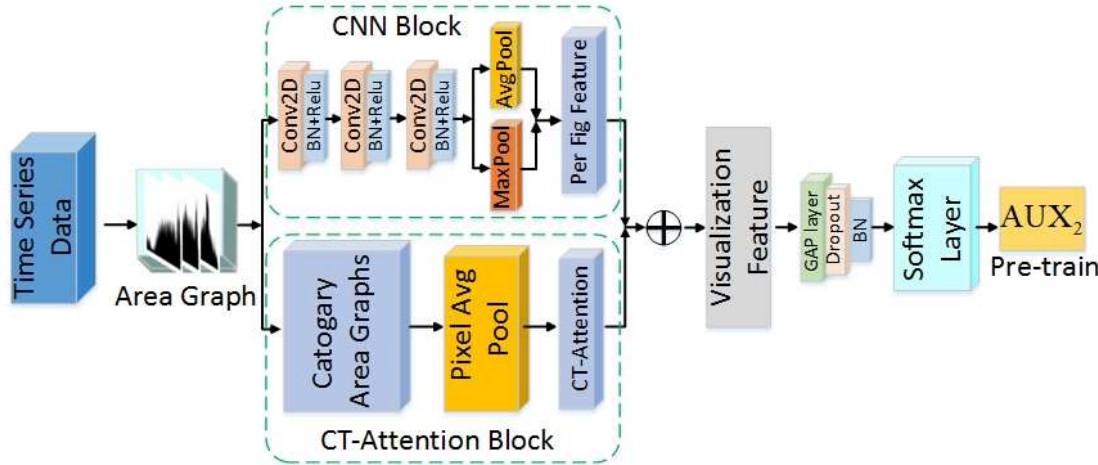

**Figure 3.** Architecture of the CNN-A model. BN stands for batch normalization, the GAP layer is the global average polling layer, and CT-Attention is the category trend attention.

Differing from the existing visualization transformation methods (e.g., Gramian fields [8,9], RP [10,11], and MTF [12]), the Area Graph is similar to human vision, directly reflecting information such as the fluctuation of TS data. The most important point is that this method is a very simple transformation method without additional calculation, avoiding the loss of some information in the calculation process. Such image data can extract visualization features through the CNN-A model.

In human vision, information such as peak value and mean value is more likely to attract attention. TS data belonging to the same category have similar peaks and averages after visualization, while data of different categories has large differences in such information. So, we used the CT-Attention block to extract these features and improve TSC. This block extracts the mean and maximum information of the category through a pixel average pooling layer based on Area Graphs of TS records belonging to same category, where the output is the CT Attention. The extracted CT attention is combined separately with the features (average and max features) extracted from each time-series record and, then, through a global average pooling layer, a Dropout layer, and a Batch Normalization layer, the final visualization features are obtained. Finally, classification is carried out through the softmax layer.

$$I = trans\,(x_i) \mid x_i \in D, \tag{4a}$$

$$C = f\,(W * I + B), \tag{4b}$$

$$H = P\,(C), \tag{4c}$$

$$CT = \prod Avg\,(Pix\,(x,y)) \mid x,y \in I, y_I = y, y \in \{1,2\cdots c\}, \tag{4d}$$

$$F^v = GAP\,(CT \odot H), \tag{4e}$$

$$P^v = softmax\,(F^v), \tag{4f}$$

$$AUX_2 = H\,(P^v, Y) = -\sum_i y_i log\,(p_i^v). \tag{4g}$$

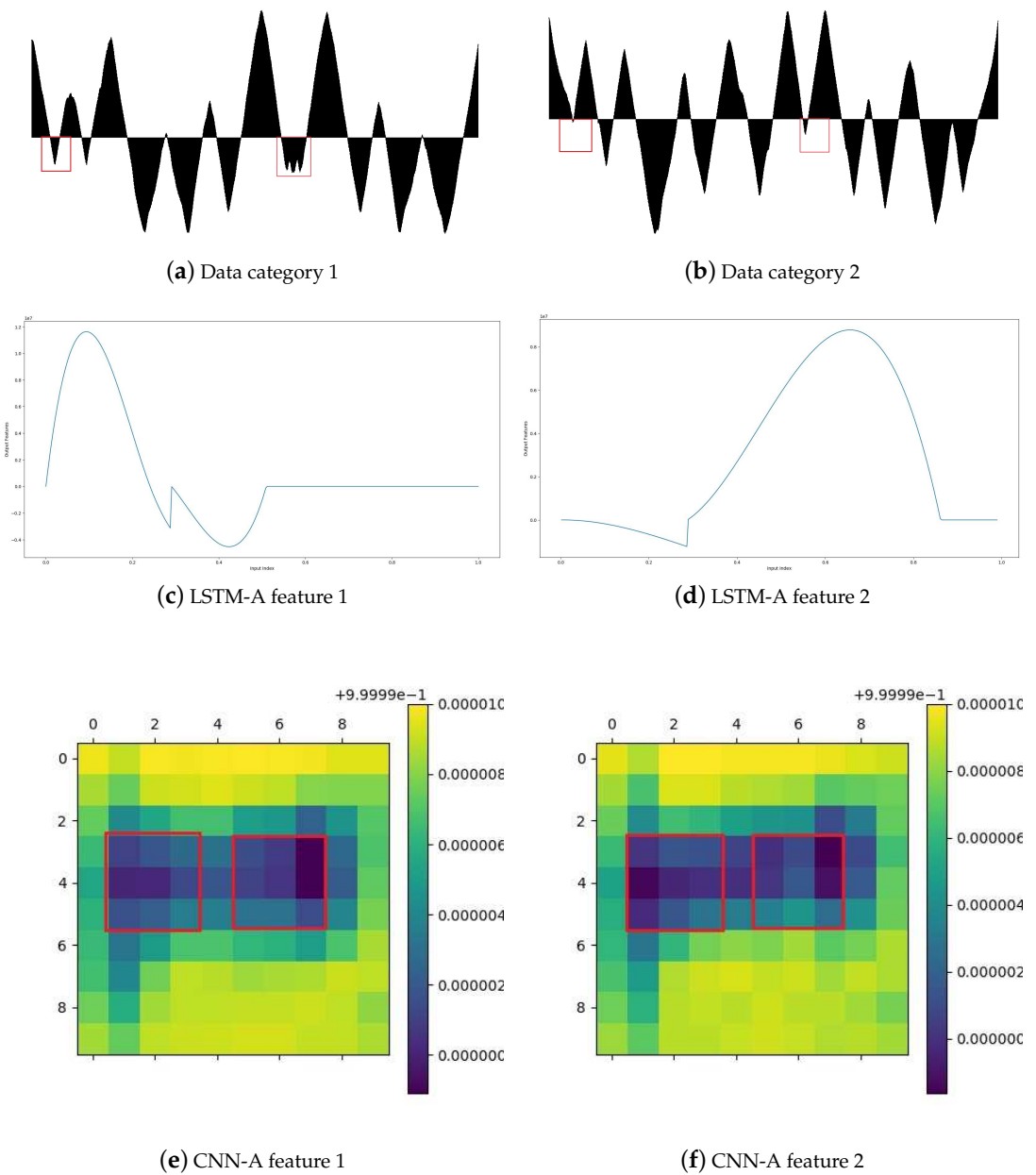

**Figure 4.** Records and features for categories of the data set BeetleFly.

The process of learning visualization features by CNN-A can be expressed by Equation (4), where *trans* means converting the TS record *x* to an Area Graph $I$, $C$ denotes the result of a convolution (where * indicates the standard dot product) applied on every $I$, $W$ and $B$ are filter parameters, and $f$ represents the combination of Batch Normalization and a final activation operation such as Rectified Linear Unit (Relu). $P$ stands for max and average pooling, and $H$ represents the pooling results. $CT$ is the category-related attention obtained by $Avg$ pooling on every pixel $(x, y)$ of Area Graphs belonging to the same category, and $y_I$ is the category that image $I$ belongs to (as shown in Figure 5). $H$ and $CT$ are merged by $\odot$ and, after $GAP$ processing, the final visualization features $F^v$ are obtained. In the pre-training phase of CNN-A, we use cross-entropy as the loss function $AUX_2$. $P^v$ is the output prediction probability sequence of CNN-A and $Y$ is the true probability sequence, and $p_i^v$ and $y_i$ are the predicted probability and true probability of the record belonging to category $i$, respectively.

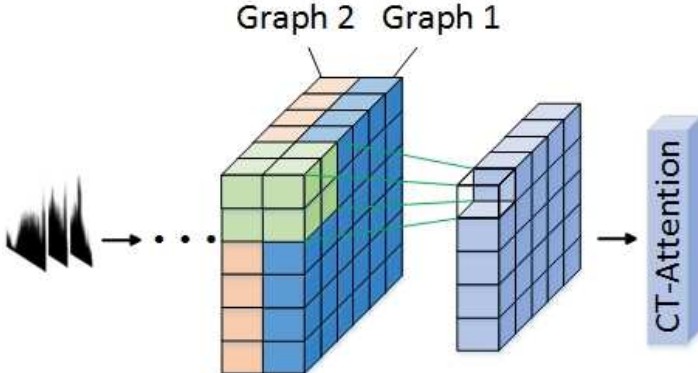

**Figure 5.** Depiction of pixel Average pooling.

## 4. Experiment

In this section, we analyze the framework from different aspects. The TS data used in our experiments was the UCR collection [15], which is the largest public TS data classification archive. The data sets in UCR involve multiple different domains and consist of clusters of different sizes, shapes, and densities. We used 112 of the data sets and excluded 15 data sets with unequal length and one (Fungi) which had a single record per category in the training data set. Through experiments, we first compared the performance of the integrated model with the selected baseline model and evaluated the factors that affected the performance of the model. Secondly, we evaluated two sub-models, including the convergence status and the effectiveness of the learned features. We also evaluated whether the attention mechanism helped to learn features better. Our source code sharing website is in Supplementary Materials.

### 4.1. Experiment Setting and Model Configuration

All experiments were conducted using Python (Keras 2.2.4 [49] and Tensorflow 1.12.0 [50] as the backend). The Python packages used for data loading, visualization, and pre-processing included Numpy 1.16.5, Pandas 0.24.2, Scikit 0.20.4, and MatPlotLib 2.2.4. The hardware information was as follows:

- CPU: Intel(R) Xeon(R) Gold 6140 CPU @ 2.30GHz (2 CPUs, 18 cores per CPU).
- GPU: NVIDIA Corporation GK210GL [Tesla K80] (8 GPUs).
- RAM: 256GB and 128GB video RAM for GPUs.
- OS: Ubuntu 18.04.3 LTS.

Table 1 shows the hyper-parameters of LSTM-A and CNN-A. Relu was selected as the activation function of the activation layer. As the basis of the proposed model is the features extracted by pre-trained LSTM-A and CNN-A from TS data, the training of the two sub-models had a great impact on feature extraction. Therefore, we adopted multiple methods to improve the training of the two sub-models in the experiments. First, we trained the two sub-models using the loss function of "categorical cross-entropy" with the Adam optimizer [51]. The initial learning rate was $1e - 3$, reducing by a factor of 0.5 to the final learning rate of $1e - 4$ every 50/100 epochs (depending on epoch sizes; for epoch sizes smaller than 100, this was set to 50, otherwise 100) of no improvement in the validation score. Second, we used the early stop method to prevent the model from overfitting. When the change range of validation accuracy was less than 0.0003% every 50 epochs, the training was stopped. Finally, we replaced the Fully-connected layer with a global average pooling layer before the softmax layer in the CNN-A model, which greatly reduced the amount of parameters. In addition, we did not perform additional pre-processing on the data (e.g., regularization), as the UCR data sets have already been z-normalized. The batch and epoch sizes were chosen from {5, 20, 100, 200} and {50, 250, 1000, 2000}, respectively. The pixel size of the Area Graphs converted from TS data was chosen from {$30 \times 30$, $150 \times 150$}.

**Table 1.** Model Configuration.

| Name | Configuration (the Parameter Value Was Selected from {·}, According to the Optimal Accuracy on the Validation Set in Different Experiments). |
|---|---|
| TSC-FF | |
| Fully-connected Layer | {64, 128, 256} |
| LSTM-A | |
| LSTM layer1 | {64, 128, 256} |
| LSTM layer2 | {64, 128, 256} |
| LSTM layer3 | {64, 128, 256} |
| Dropout | {0.2, 0.5, 0.8} |
| Batch Normalization | momentum = 0.99, epsilon = 0.001 |
| Fully-connected Layer | {64, 128, 256} |
| CNN-A | |
| Conv2D1 | filter number {32, 64, 128}, kernel size {5 × 5, 8 × 8, 10 × 10}, strides {1, 2, 4} |
| Conv2D2 | filter number {32, 64, 128}, kernel size {5 × 5, 8 × 8, 10 × 10}, strides {1, 2, 4} |
| Conv2D3 | filter number {32, 64, 128}, kernel size {5 × 5, 8 × 8, 10 × 10}, strides {1, 2, 4} |
| Dropout | {0.2, 0.5, 0.8} |
| Batch Normalization | momentum = 0.99, epsilon = 0.001 |
| Avg. Pool | pool size {5 × 5, 8 × 8} |
| Max Pool | pool size {5 × 5, 8 × 8} |
| Fully-connected Layer | {64, 128, 256} |

In the field of TSC research, much work based on LSTM, CNN [27,52], and hybrid LSTM-CNN [39–41] has been carried out, achieving promising results. The architectures of these models are used for reference. In the experiments, we also tried models with different LSTM layers or CNN layers. We found that reducing the number of layers damaged the performance of model, while increasing the number of layers not only damaged the performance of model as well, but also increased the model training time.

### 4.2. Integrated Model Evaluation

Among the existing four types of methods, we used 15 methods for comparison, among which we reproduced 1NN_DTW, ST and FCN, supplementing the results on the 27 data sets of the additional data sets in UCR, and the results of the remaining methods come from [15]. The chosen methods were as follows:

- Distance-based methods: 1NN_DTW [2] and PF [17];
- Feature-based methods: BOSS [20], ST [18,19], STC [24], WEASEL [23], TSF [21] and RISE [25];
- Ensemble methods: HIVE-COTE [4], TS-CHIEF [34], ROCKET [35] and Catch22 [36];
- Deep learning methods: ResNet [38], FCN [38] and InceptionTime [42].

For the detailed structure and parameters of each method, please refer to their respective references. All results were obtained by taking the average value from multiple experiments. All models were evaluated using classification accuracy and mean-per-class-error (MPCE), which is defined as the average error of each class for all data sets and mathematically represented in Equation (5). The Average Arithmetic Rank (AVG Rank) is the mean of the classification accuracy ranking over the 112 data sets. Table 2 lists the comparison results. In order to facilitate the display, the names of some methods have been abbreviated again, including 1NN_DTW (labelled DTW), WEASEL (WS), HIVE-COTE (HCT), TS-CHIEF (CHI), ROCKET (RK), Catch22 (C2), ResNet (RN), InceptionTime (IcT).

$$PCE_k = \frac{1 - accuracy}{number\ of\ classes}, \tag{5a}$$

$$MPCE = \frac{1}{N} \sum_{k=1}^{N} PCE_k. \tag{5b}$$

Compared with the baseline methods on 112 data sets in UCR, the proposed mthod TSC-FF won or tied on 20 data sets. Judging from the AVG Rank and MPSE, its overall ranking was fifth, after HIVE-COTE, TS-CHIEF, ROCKET and InceptionTime. By comparison, we found that TSC-FF was far better than six feature-based methods. Therefore, even though the deep learning-based method TSC-FF proposed in this paper did not achieve the best performance, it showed the greatest potential of the deep learning methods considered for feature learning, proving that DNN can automatically complete feature extraction, not requiring complicated feature engineering. With the development of DNN and the accumulation of data, deep learning methods can even achieve better performance.

Although TSC-FF could not defeat the deep learning method InceptionTime, which achieved the lowest MPSE as a whole, by comparing the results of FordA, FordB, and Wafer, we found that our framework achieved higher accuracy on these data sets. This shows that, when a DNN is given enough data to learn features, using pre-trained sub-models to learn features from different aspects and then fusing features can improve DNN performance. Of course, the amount of data and the number of categories also has an effect on DNN performance, which will be explored in later experiments. On the data set ElectricDevices, the featured-based method BOSS had the highest accuracy. The reason for this is the artificially constructed features used by BOSS are better than those learned by the DNN. After all, there still exists a certain gap between DNNs and the human brain. On some data sets, the features learned by DNN are not as effective as those designed by humans. However, the results of BOSS on other data sets proved the weak generality of artificially constructed features.

**Table 2.** Accuracy comprison on 112 data sets. Bold values denote the model with the best performance.

| | DTW | PF | BOSS | ST | STC | WS | TSF | RISE | HCT | CHI | RK | C2 | RN | FCN | IcT | TSC-FF |
|---|---|---|---|---|---|---|---|---|---|---|---|---|---|---|---|---|
| ACSF1 | 0.5600 | 0.6383 | 0.7683 | 0.7826 | 0.8383 | 0.8180 | 0.6350 | 0.7600 | 0.8500 | 0.8070 | 0.8070 | 0.7777 | 0.8240 | **0.8800** | 0.8267 | 0.8552 |
| Adiac | 0.5857 | 0.7222 | 0.7490 | 0.7826 | 0.7932 | 0.7988 | 0.7119 | 0.7580 | 0.7962 | 0.7797 | 0.7720 | 0.6847 | 0.8154 | **0.8570** | 0.8223 | 0.7875 |
| ArroHea | 0.7771 | **0.8836** | 0.8688 | 0.7371 | 0.8067 | 0.8484 | 0.7968 | 0.8282 | 0.8760 | 0.8811 | 0.8590 | 0.7503 | 0.8587 | 0.8800 | 0.8804 | 0.8025 |
| Beef | 0.6667 | 0.5944 | 0.6122 | 0.9000 | 0.7356 | 0.7400 | 0.6889 | 0.7422 | 0.7356 | 0.6322 | 0.7600 | 0.4733 | 0.6767 | 0.7500 | 0.6822 | **0.8444** |
| BeetleFl | 0.7500 | 0.8600 | 0.9433 | 0.9000 | 0.9333 | 0.8867 | 0.8333 | 0.8717 | **0.9633** | 0.9583 | 0.8850 | 0.8400 | 0.8533 | 0.9500 | 0.8933 | 0.8889 |
| BirChick | 0.8500 | 0.9033 | **0.9833** | 0.8000 | 0.8700 | 0.8650 | 0.8150 | 0.8683 | 0.9400 | 0.9633 | 0.8817 | 0.8933 | 0.9450 | 0.9500 | 0.9517 | 0.9222 |
| BME | 0.7933 | **0.9991** | 0.8658 | 0.8267 | 0.9298 | 0.9478 | 0.9624 | 0.7860 | 0.9822 | 0.9964 | 0.9973 | 0.9049 | **0.9991** | 0.7533 | 0.9964 | 0.8921 |
| Car | 0.7333 | 0.8056 | 0.8483 | 0.9167 | 0.8583 | 0.8344 | 0.7661 | 0.7533 | 0.8689 | 0.8789 | 0.9117 | 0.7461 | 0.9083 | 0.9170 | 0.9011 | **0.9500** |
| CBF | 0.5922 | 0.9936 | 0.9989 | 0.9744 | 0.9853 | 0.9798 | 0.9719 | 0.9490 | 0.9983 | 0.9984 | 0.9959 | 0.9537 | 0.9882 | **1.0000** | 0.9961 | **1.0000** |
| Chinatow | 0.8406 | 0.9480 | 0.8771 | 0.9145 | 0.9630 | 0.9573 | 0.9530 | 0.8885 | 0.9628 | 0.9618 | 0.9669 | 0.9345 | 0.9701 | 0.9826 | 0.9649 | **0.9861** |
| ChlCon | 0.6831 | 0.6311 | 0.6582 | 0.6997 | 0.7352 | 0.7549 | 0.7231 | 0.7648 | 0.7339 | 0.6608 | 0.7961 | 0.5980 | 0.8410 | 0.8430 | **0.8636** | 0.7246 |
| CCECGTor | 0.7109 | 0.9377 | 0.9148 | 0.9543 | 0.9778 | 0.9850 | 0.9582 | 0.9474 | **0.9937** | 0.9534 | 0.8641 | 0.8030 | 0.7679 | 0.8130 | 0.8328 | 0.9511 |
| Coffee | 0.9286 | 0.9917 | 0.9857 | 0.9643 | 0.9893 | 0.9893 | 0.9869 | 0.9845 | 0.9929 | 0.9905 | 1.0000 | 0.9798 | 0.9964 | **1.0000** | 0.9988 | **1.0000** |
| Compu | 0.6680 | 0.7143 | 0.8005 | 0.7360 | 0.7991 | 0.7785 | 0.6488 | 0.7789 | 0.8111 | 0.7539 | 0.8429 | 0.7803 | 0.8604 | 0.8480 | **0.8656** | 0.7200 |
| CricketX | 0.6308 | 0.8004 | 0.7624 | 0.7718 | 0.7921 | 0.7757 | 0.6927 | 0.7062 | 0.8162 | 0.8304 | 0.8390 | 0.6089 | 0.8081 | 0.8150 | **0.8532** | 0.8077 |
| CricketY | 0.5590 | 0.7998 | 0.7504 | 0.7795 | 0.7779 | 0.7796 | 0.6859 | 0.7091 | 0.8098 | 0.8170 | 0.8450 | 0.5905 | 0.8102 | 0.7920 | **0.8600** | 0.8000 |
| CricketZ | 0.5564 | 0.8028 | 0.7693 | 0.7872 | 0.8072 | 0.7899 | 0.7059 | 0.7216 | 0.8339 | 0.8382 | 0.8532 | 0.6282 | 0.8132 | 0.8130 | **0.8611** | 0.8282 |
| Crop | 0.6488 | 0.7531 | 0.6857 | 0.7089 | 0.7367 | 0.7238 | 0.7456 | 0.7300 | 0.7682 | 0.7621 | 0.7517 | 0.6531 | 0.7638 | 0.7537 | **0.7932** | 0.7622 |
| DiaSizRed | 0.9346 | 0.9568 | 0.9452 | 0.9248 | 0.8593 | 0.9081 | 0.9417 | 0.9321 | 0.9143 | 0.9459 | **0.9580** | 0.9247 | 0.3062 | 0.9300 | 0.9507 | 0.9471 |
| DiPhOuAG | 0.7319 | 0.8022 | 0.8206 | 0.7754 | 0.7962 | 0.7928 | 0.8094 | 0.8216 | 0.8240 | 0.8281 | 0.8115 | 0.7830 | 0.7760 | **0.8350** | 0.7657 | 0.7661 |
| DiPhOuCor | 0.6763 | 0.8234 | 0.8117 | 0.7698 | **0.8273** | 0.8192 | 0.8058 | 0.8112 | 0.8236 | 0.8193 | 0.8244 | 0.8121 | 0.8092 | 0.8120 | 0.8156 | 0.7120 |
| DiPhTW | 0.5899 | 0.6921 | 0.6715 | 0.6619 | 0.6899 | 0.6787 | 0.6911 | 0.6945 | 0.6962 | 0.6918 | 0.7012 | 0.6811 | 0.6671 | **0.7900** | 0.6659 | 0.6875 |
| Earthqu | 0.6763 | 0.7496 | 0.7460 | 0.7410 | 0.7420 | 0.7475 | 0.7475 | 0.7482 | 0.7475 | 0.7482 | 0.7484 | 0.7388 | 0.7170 | 0.8010 | 0.7321 | **0.8160** |
| ECG200 | 0.8300 | 0.8730 | 0.8783 | 0.8300 | 0.8390 | 0.8590 | 0.8600 | 0.8510 | 0.8587 | 0.8550 | 0.8990 | 0.7887 | 0.8837 | **0.9000** | 0.8957 | 0.8916 |
| ECG5000 | 0.9196 | 0.9395 | 0.9401 | 0.9438 | 0.9418 | 0.9459 | 0.9434 | 0.9367 | 0.9456 | **0.9485** | 0.9474 | 0.9362 | 0.9370 | 0.9410 | 0.9421 | 0.9481 |
| ECGFiveD | 0.6864 | 0.8828 | 0.9923 | 0.9837 | 0.9779 | 0.9935 | 0.9520 | 0.9729 | 0.9938 | 0.9944 | 0.9960 | 0.8159 | 0.9510 | 0.9850 | 0.9959 | **0.9991** |
| ElecDev | 0.5709 | 0.8424 | 0.7974 | 0.7470 | 0.8818 | 0.8717 | 0.7931 | 0.8240 | 0.8797 | 0.8650 | 0.8932 | 0.8721 | 0.8884 | 0.7230 | **0.8902** | 0.7413 |
| EOGHoSi | 0.5470 | 0.8244 | 0.7067 | 0.7761 | 0.7599 | 0.7473 | 0.7060 | 0.6249 | 0.7957 | 0.8537 | 0.8138 | 0.6730 | 0.8647 | 0.7298 | **0.8836** | 0.8358 |
| EOGVeSi | 0.5801 | 0.7776 | 0.6603 | 0.7515 | 0.7132 | 0.6975 | 0.6726 | 0.6035 | 0.7630 | 0.8095 | 0.7811 | 0.6149 | 0.7517 | 0.7254 | **0.8145** | 0.7722 |
| EthLevel | 0.2680 | 0.3241 | 0.5087 | 0.6256 | 0.8557 | 0.7051 | 0.6693 | 0.6611 | 0.8490 | 0.6056 | 0.6253 | 0.3974 | 0.8525 | 0.6420 | **0.8755** | 0.8442 |
| FacAll | 0.8728 | 0.9771 | 0.9701 | 0.7787 | 0.9537 | 0.9731 | 0.9500 | 0.9649 | 0.9797 | 0.9827 | **0.9880** | 0.8108 | 0.9819 | 0.9290 | 0.9833 | 0.8705 |
| FaFour | 0.6250 | 0.9455 | 0.9955 | 0.8523 | 0.6564 | 0.9811 | 0.9068 | 0.8769 | 0.9731 | **0.9996** | 0.9314 | 0.6795 | 0.9250 | 0.9320 | 0.9386 | 0.9091 |
| FacUCR | 0.8434 | 0.9561 | 0.9513 | 0.9059 | 0.9101 | 0.9562 | 0.9040 | 0.8920 | 0.9613 | 0.9729 | 0.9713 | 0.7087 | 0.9643 | 0.9480 | **0.9769** | 0.9593 |
| FiftyW | 0.6923 | 0.8259 | 0.7059 | 0.7055 | 0.7366 | 0.7770 | 0.7217 | 0.6666 | 0.7716 | **0.8427** | 0.8251 | 0.5978 | 0.7240 | 0.6790 | 0.8268 | 0.8041 |

**Table 2.** *Cont.*

| | DTW | PF | BOSS | ST | STC | WS | TSF | RISE | HCT | CHI | RK | C2 | RN | FCN | IcT | TSC-FF |
|---|---|---|---|---|---|---|---|---|---|---|---|---|---|---|---|---|
| Fish | 0.8971 | 0.9339 | 0.9697 | **0.9886** | 0.9499 | 0.9509 | 0.8305 | 0.8590 | 0.9794 | 0.9815 | 0.9741 | 0.7726 | 0.9699 | 0.9710 | 0.9726 | **0.9886** |
| FordA | 0.7174 | 0.8500 | 0.9214 | 0.9712 | 0.9342 | 0.9687 | 0.8158 | 0.9400 | 0.9441 | 0.9474 | 0.9424 | 0.9092 | 0.9315 | 0.9060 | 0.9591 | **1.0000** |
| FordB | 0.6654 | 0.8394 | 0.9074 | 0.8074 | 0.9198 | 0.9371 | 0.7907 | 0.9173 | 0.9280 | 0.9195 | 0.9230 | 0.8701 | 0.9128 | 0.8830 | 0.9409 | **1.0000** |
| FrRegTra | 0.9046 | 0.9424 | 0.9881 | 0.9951 | **0.9993** | 0.9906 | 0.9971 | 0.9523 | 0.9991 | 0.9985 | 0.9944 | 0.9982 | 0.9967 | 0.9965 | 0.9958 | 0.9989 |
| FrSmaTra | 0.7147 | 0.8234 | 0.9616 | 0.9689 | 0.9886 | 0.9006 | 0.9614 | 0.8787 | 0.9837 | **0.9955** | 0.9872 | 0.9598 | 0.9495 | 0.6902 | 0.9489 | 0.9668 |
| GunPoint | 0.9933 | 0.9913 | 0.9964 | **1.0000** | 0.9864 | 0.9931 | 0.9553 | 0.9809 | 0.9982 | **1.0000** | 0.9920 | 0.9431 | 0.9909 | **1.0000** | 0.9951 | **1.0000** |
| GPAgSpa | 0.9505 | 0.9969 | 0.9949 | 0.9805 | 0.9660 | 0.9813 | 0.9777 | 0.9863 | 0.9966 | 0.9996 | 0.9935 | 0.9439 | 0.9944 | 0.9424 | 0.9839 | **1.0000** |
| GPMaVeFe | 0.9468 | 0.9994 | 0.9996 | 0.9915 | 0.9865 | 0.9939 | 0.9960 | 0.9911 | 0.9999 | 0.9994 | 0.9999 | 0.9935 | 0.9901 | 0.9494 | **0.9983** | 0.9957 |
| GPOlVeYo | 0.8560 | **1.0000** | 0.9992 | 0.9945 | 0.9783 | 0.9860 | **1.0000** | 0.9998 | **1.0000** | **1.0000** | 0.9896 | 0.9642 | **1.0000** | 0.8952 | **1.0000** | 0.9264 |
| Ham | 0.4762 | 0.7835 | 0.8375 | 0.6857 | 0.8108 | 0.8213 | 0.7994 | 0.8197 | 0.8400 | 0.8051 | 0.8552 | 0.6940 | 0.8073 | 0.7620 | **0.8502** | 0.7128 |
| HandOut | 0.7378 | N/A | 0.9148 | 0.9324 | 0.9214 | **0.9454** | 0.9066 | 0.8810 | 0.9166 | N/A | N/A | 0.8684 | 0.9030 | 0.7760 | N/A | 0.9069 |
| Haptics | 0.2857 | 0.4583 | 0.4674 | 0.5227 | 0.5417 | 0.4492 | 0.4656 | 0.4788 | 0.5394 | 0.5233 | 0.5342 | 0.4587 | 0.4965 | **0.5510** | 0.5366 | 0.5213 |
| Herring | 0.5000 | 0.5745 | 0.5958 | 0.6719 | 0.6328 | 0.6021 | 0.6042 | 0.5984 | 0.6120 | 0.5974 | 0.6250 | 0.5557 | 0.5969 | **0.7030** | 0.6250 | 0.6491 |
| HouTwe | 0.8571 | 0.9370 | 0.9560 | 0.9204 | 0.9751 | 0.8106 | 0.8378 | 0.9297 | **0.9787** | 0.9703 | 0.9627 | 0.9462 | 0.9555 | 0.8908 | 0.9535 | 0.9613 |
| InlSkate | 0.4527 | 0.5610 | 0.5056 | 0.3727 | 0.4393 | **0.6746** | 0.3728 | 0.3904 | 0.5152 | 0.5719 | 0.4379 | 0.4724 | 0.4101 | 0.4110 | 0.5344 | 0.4616 |
| IERegTra | **1.0000** | **1.0000** | 0.9968 | 0.9957 | 0.9934 | 0.9718 | **1.0000** | 0.9929 | **1.0000** | **1.0000** | 0.9870 | 0.9288 | **1.0000** | **1.0000** | **1.0000** | **1.0000** |
| IESmaTra | **1.0000** | **1.0000** | 0.9849 | 0.8736 | 0.8877 | 0.9067 | 0.9959 | 0.9722 | **1.0000** | **1.0000** | 0.9343 | 0.8372 | 0.3963 | 0.3574 | **1.0000** | 0.5150 |
| InWinSou | 0.2434 | 0.6069 | 0.5118 | 0.6268 | 0.6286 | 0.6195 | 0.6035 | 0.6364 | 0.6403 | 0.6322 | **0.6567** | 0.5592 | 0.4914 | 0.4020 | 0.6273 | 0.6269 |
| IPoDem | 0.9135 | 0.9560 | 0.8709 | 0.9475 | 0.9538 | 0.9468 | 0.9595 | 0.9445 | 0.9582 | 0.9624 | 0.9616 | 0.8775 | 0.9571 | **0.9700** | 0.9603 | 0.9633 |
| LarKitApp | 0.7680 | 0.8101 | 0.8358 | 0.8587 | 0.9299 | 0.7927 | 0.6363 | 0.7358 | 0.9201 | 0.8602 | 0.9300 | 0.8863 | **0.9539** | 0.8960 | 0.9524 | 0.8723 |
| Lightn2 | 0.6721 | **0.8486** | 0.8191 | 0.7377 | 0.6585 | 0.6273 | 0.7645 | 0.6820 | 0.7732 | 0.7689 | 0.7765 | 0.7448 | 0.8005 | 0.8030 | 0.8164 | 0.8155 |
| Lightn7 | 0.5753 | 0.7922 | 0.6712 | 0.7260 | 0.7434 | 0.7128 | 0.7205 | 0.6977 | 0.7575 | 0.7936 | 0.7977 | 0.6461 | 0.8100 | **0.8630** | 0.8215 | 0.8115 |
| Mallat | 0.8972 | 0.9720 | 0.9498 | 0.9642 | 0.9080 | 0.9664 | 0.9357 | 0.9541 | 0.9672 | 0.9767 | 0.9572 | 0.9061 | 0.9690 | **0.9800** | 0.9625 | 0.9542 |
| Meat | 0.6667 | 0.9872 | 0.9806 | 0.8500 | 0.9678 | 0.9767 | 0.9839 | 0.9867 | 0.9861 | 0.9844 | 0.9889 | 0.9428 | **0.9939** | 0.9670 | 0.9839 | 0.9333 |
| MedIma | 0.6632 | 0.7714 | 0.7158 | 0.6697 | 0.7098 | 0.7088 | 0.7462 | 0.6667 | 0.7404 | 0.7991 | **0.8051** | 0.7568 | 0.7922 | 0.7920 | 0.7963 | 0.7578 |
| MiPhOC | 0.6907 | 0.6589 | 0.6563 | 0.7938 | 0.6675 | 0.6604 | 0.6595 | 0.6998 | 0.6978 | 0.6944 | 0.7108 | 0.6881 | 0.5965 | **0.7680** | 0.5946 | 0.7663 |
| MiPhAG | 0.5584 | 0.8241 | 0.8095 | 0.6429 | 0.8315 | 0.8283 | 0.7995 | 0.8055 | 0.8129 | 0.8058 | **0.8345** | 0.7727 | 0.8236 | 0.7950 | 0.8340 | 0.7000 |
| MiPhTW | 0.5000 | 0.5491 | 0.5323 | 0.5195 | 0.5786 | 0.5543 | 0.5686 | 0.5851 | 0.5838 | 0.5732 | 0.5898 | 0.5574 | 0.5312 | **0.6120** | 0.5266 | 0.5745 |
| MSRegTra | 0.7518 | 0.9638 | 0.9179 | 0.9381 | 0.9602 | 0.9633 | 0.9230 | 0.9377 | 0.9659 | **0.9714** | 0.9657 | 0.9215 | 0.9701 | 0.8668 | 0.9663 | 0.9690 |
| MSSmaTra | 0.8043 | 0.9281 | 0.8717 | 0.9061 | 0.9368 | 0.9245 | 0.8402 | 0.8983 | 0.9447 | **0.9473** | 0.9307 | 0.8752 | 0.9162 | 0.9105 | 0.9134 | 0.9386 |
| MotStra | 0.7157 | 0.9150 | 0.8442 | 0.8970 | 0.9089 | 0.9048 | 0.8555 | 0.8780 | 0.9365 | 0.9301 | 0.9055 | 0.8485 | 0.9031 | **0.9500** | 0.8806 | 0.8943 |
| NonECG1 | 0.6972 | 0.8789 | 0.8416 | 0.9496 | 0.9359 | 0.9296 | 0.8848 | 0.9007 | 0.9278 | 0.9167 | 0.9022 | 0.8481 | 0.9442 | **0.9610** | 0.8744 | 0.8818 |
| NonECG2 | 0.8265 | 0.8585 | 0.9027 | 0.9511 | 0.9501 | 0.9362 | 0.9156 | 0.9184 | 0.9507 | **0.9736** | 0.9387 | 0.8772 | 0.9501 | 0.9550 | 0.9526 | 0.9424 |
| OliveOil | 0.8667 | 0.8291 | 0.8756 | **0.9000** | 0.8789 | 0.9133 | 0.8933 | 0.8933 | 0.8833 | 0.8253 | 0.8447 | 0.7456 | 0.8622 | 0.8330 | 0.8615 | 0.8667 |
| OSULeaf | 0.8802 | 0.3204 | 0.9691 | 0.9669 | 0.9563 | 0.8518 | 0.6433 | 0.6543 | 0.9748 | 0.3698 | 0.2751 | 0.7242 | 0.9747 | **0.9880** | 0.3346 | 0.9699 |

**Table 2.** *Cont.*

| | DTW | PF | BOSS | ST | STC | WS | TSF | RISE | HCT | CHI | RK | C2 | RN | FCN | IcT | TSC-FF |
|---|---|---|---|---|---|---|---|---|---|---|---|---|---|---|---|---|
| PhOutCor | 0.7401 | 0.2755 | 0.8174 | 0.7634 | 0.8337 | 0.8217 | 0.8057 | 0.8125 | 0.8265 | **0.9601** | 0.1955 | 0.7919 | 0.8477 | 0.8260 | 0.9221 | 0.8096 |
| Phoneme | 0.2563 | 0.7157 | 0.2546 | 0.3207 | 0.3565 | 0.2595 | 0.1937 | 0.3466 | 0.3712 | **0.9689** | 0.9085 | 0.3002 | 0.3457 | 0.3450 | 0.9332 | 0.3058 |
| PigAir | 0.2260 | N/A | 0.9418 | 0.9656 | **0.9774** | 0.5378 | 0.2801 | 0.1825 | 0.9577 | N/A | N/A | 0.2769 | 0.4370 | 0.2067 | N/A | 0.5370 |
| PigArt | 0.6538 | N/A | 0.9676 | 0.9584 | 0.9457 | 0.9378 | 0.3072 | 0.7946 | 0.9668 | N/A | N/A | 0.8236 | 0.9537 | 0.8798 | N/A | **0.9719** |
| PigCVP | 0.4231 | 0.5005 | 0.9595 | 0.9291 | 0.8997 | 0.8979 | 0.1782 | 0.6825 | 0.9530 | **0.9614** | 0.8817 | 0.4761 | 0.4540 | 0.2500 | 0.9053 | 0.5253 |
| Plane | **1.0000** | **1.0000** | 0.9981 | **1.0000** | 0.9990 | 0.9949 | 0.9959 | 0.9965 | **1.0000** | **1.0000** | **1.0000** | 0.9883 | **1.0000** | **1.0000** | 0.9968 | **1.0000** |
| PowCons | 0.7611 | 0.9874 | 0.8900 | 0.8906 | 0.9406 | 0.9194 | **0.9931** | 0.9580 | 0.9924 | 0.9794 | 0.9561 | 0.8863 | 0.8861 | 0.8389 | 0.9861 | 0.9601 |
| PPOAG | 0.8179 | 0.8402 | 0.8285 | **0.8832** | 0.8455 | 0.8449 | 0.8450 | 0.8572 | 0.8559 | 0.8463 | 0.8524 | 0.8584 | 0.8174 | 0.8490 | 0.8221 | 0.8538 |
| PPOCor | 0.8195 | 0.8659 | 0.8655 | 0.8439 | 0.8953 | 0.8763 | 0.8489 | 0.8737 | 0.8852 | 0.8755 | 0.8993 | 0.8337 | 0.9056 | 0.9000 | **0.9063** | 0.8761 |
| ProPhTW | 0.7512 | 0.7907 | 0.7686 | 0.8049 | 0.8076 | 0.8015 | 0.8016 | 0.8132 | **0.8161** | 0.8111 | 0.8036 | 0.7863 | 0.7894 | 0.8100 | 0.7807 | 0.8043 |
| RefDev | 0.4080 | 0.6724 | 0.7828 | 0.5813 | 0.7772 | 0.7397 | 0.6116 | 0.6518 | **0.7909** | 0.7268 | 0.7300 | 0.7088 | 0.7868 | 0.5330 | 0.7594 | 0.5697 |
| Rock | 0.6200 | 0.7747 | 0.8027 | 0.8525 | **0.8673** | 0.8547 | 0.7600 | 0.7820 | 0.8553 | 0.8320 | 0.8047 | 0.7053 | 0.4160 | 0.3800 | 0.6273 | 0.6133 |
| ScreTyp | 0.4267 | 0.5720 | 0.5848 | 0.5200 | 0.7179 | 0.5959 | 0.4718 | 0.6059 | 0.7242 | 0.5942 | 0.6090 | 0.6007 | **0.7553** | 0.6670 | 0.7056 | 0.5392 |
| SeHaGen | 0.7450 | 0.9631 | 0.8877 | 0.8464 | 0.9323 | 0.7814 | 0.9474 | 0.8700 | **0.9692** | 0.9389 | 0.9231 | 0.8706 | 0.8089 | 0.7433 | 0.8846 | 0.9521 |
| SeHaMov | 0.6600 | **0.8909** | 0.6649 | 0.7094 | 0.7883 | 0.4603 | 0.8641 | 0.7240 | 0.8890 | 0.8850 | 0.6526 | 0.6825 | 0.4210 | 0.4200 | 0.5510 | 0.5916 |
| SeHaSub | 0.7022 | 0.9381 | 0.8351 | 0.8369 | 0.9133 | 0.7935 | 0.9160 | 0.8241 | **0.9506** | 0.9331 | 0.9119 | 0.8487 | 0.5547 | 0.5711 | 0.7639 | 0.6121 |
| ShapSim | 0.5389 | 0.7893 | **1.0000** | 0.9556 | 0.9996 | 0.9974 | 0.5137 | 0.7676 | **1.0000** | **1.0000** | 0.9981 | 0.9937 | 0.7272 | 0.8670 | 0.9235 | 0.8062 |
| ShapAll | 0.8350 | 0.8901 | 0.9085 | 0.8417 | 0.8653 | 0.9160 | 0.8043 | 0.8501 | 0.9313 | 0.9411 | 0.9232 | 0.8293 | 0.9331 | 0.8980 | **0.9382** | 0.8148 |
| SmKApp | 0.6533 | 0.7381 | 0.7471 | 0.7920 | 0.8128 | 0.8052 | 0.7882 | 0.8060 | 0.8283 | **0.8380** | 0.8183 | 0.8170 | 0.7966 | 0.8030 | 0.7706 | 0.8146 |
| SmoSub | 0.7867 | **0.9980** | 0.4073 | 0.7800 | 0.9358 | 0.8553 | 0.9873 | 0.8484 | 0.9862 | 0.9973 | 0.9749 | 0.8553 | 0.9929 | 0.8467 | 0.9849 | 0.9833 |
| SonSur1 | 0.7304 | 0.9201 | 0.8977 | 0.8436 | 0.8009 | 0.9093 | 0.8637 | 0.8670 | 0.8263 | 0.8897 | 0.9581 | 0.8834 | 0.9604 | **0.9680** | 0.9542 | 0.7630 |
| SonSur2 | 0.8583 | 0.8990 | 0.8794 | 0.9339 | 0.9370 | 0.9353 | 0.8743 | 0.9125 | 0.9367 | 0.9011 | 0.9350 | 0.9023 | **0.9689** | 0.9620 | 0.9513 | 0.9016 |
| StaCurv | 0.9622 | 0.9796 | 0.9766 | 0.9785 | 0.9786 | 0.9804 | 0.9704 | 0.9744 | 0.9796 | **0.9811** | 0.9810 | 0.9702 | 0.9720 | 0.9670 | 0.9781 | 0.9799 |
| Strberry | 0.9622 | 0.9605 | 0.9705 | 0.9622 | 0.9717 | 0.9786 | 0.9675 | 0.9730 | 0.9751 | 0.9740 | **0.9787** | 0.9229 | 0.9749 | 0.9690 | 0.9755 | 0.9709 |
| SweLeaf | 0.8848 | 0.9531 | 0.9202 | 0.9280 | 0.9340 | 0.9576 | 0.8979 | 0.9229 | 0.9494 | 0.9619 | 0.9632 | 0.8805 | 0.9586 | 0.9660 | **0.9698** | 0.8779 |
| Symbols | **0.9709** | 0.9670 | 0.9632 | 0.8824 | 0.9014 | 0.9532 | 0.8779 | 0.9127 | 0.9685 | 0.9708 | 0.9686 | 0.9480 | 0.9467 | 0.9620 | 0.9695 | 0.8000 |
| SynCont | 0.5667 | 0.9983 | 0.9666 | 0.9833 | 0.9919 | 0.9870 | 0.9916 | 0.6779 | 0.9942 | 0.9990 | 0.9978 | 0.9670 | 0.9944 | 0.9900 | 0.9958 | **1.0000** |
| ToeSeg1 | 0.7851 | 0.8355 | 0.9249 | 0.9649 | 0.9534 | 0.9430 | 0.6671 | 0.8804 | 0.9596 | 0.9598 | 0.9329 | 0.8127 | 0.9542 | 0.9690 | 0.9532 | **0.9793** |
| ToeSeg2 | 0.6846 | 0.8859 | 0.9615 | 0.9077 | 0.9451 | 0.9285 | 0.8026 | 0.9118 | **0.9682** | 0.9626 | 0.9326 | 0.8351 | 0.9531 | 0.9150 | 0.9636 | 0.9318 |
| Trace | **1.0000** | **1.0000** | **1.0000** | **1.0000** | **1.0000** | **1.0000** | 0.9920 | 0.9830 | **1.0000** | **1.0000** | **1.0000** | 0.9997 | **1.0000** | **1.0000** | **1.0000** | **1.0000** |
| TwLeECG | 0.9947 | 0.9818 | 0.9847 | 0.9974 | 0.9991 | 0.9975 | 0.8706 | 0.9107 | 0.9963 | 0.9901 | 0.9985 | 0.8539 | 0.9994 | **1.0000** | 0.9948 | 0.9288 |
| TwoPatt | 0.9975 | **1.0000** | 0.9917 | 0.9550 | 0.9903 | 0.9814 | 0.9938 | 0.4390 | 0.9995 | **1.0000** | **1.0000** | 0.8486 | 0.9998 | 0.8970 | **1.0000** | **1.0000** |

**Table 2.** *Cont.*

| | DTW | PF | BOSS | ST | STC | WS | TSF | RISE | HCT | CHI | RK | C2 | RN | FCN | IcT | TSC-FF |
|---|---|---|---|---|---|---|---|---|---|---|---|---|---|---|---|---|
| UMD | 0.8472 | 0.9544 | 0.9664 | 0.9514 | 0.9368 | 0.9322 | 0.8333 | 0.5412 | 0.9674 | 0.9833 | 0.9829 | 0.8694 | 0.9525 | 0.9458 | **0.9799** | 0.8885 |
| UWavAll | 0.6770 | 0.9731 | 0.9449 | 0.8029 | 0.9580 | 0.9578 | 0.9624 | 0.9174 | 0.9666 | 0.9707 | **0.9773** | 0.8264 | 0.8690 | 0.8260 | 0.9512 | 0.9705 |
| UWavX | 0.5840 | 0.8314 | 0.7526 | 0.7303 | 0.8202 | 0.8176 | 0.8000 | 0.6339 | 0.8335 | 0.8468 | **0.8571** | 0.7685 | 0.7904 | 0.7540 | 0.8336 | 0.8460 |
| UWavY | 0.5771 | 0.7672 | 0.6621 | 0.7485 | 0.7449 | 0.7258 | 0.7219 | 0.6682 | 0.7547 | **0.7880** | 0.7836 | 0.7044 | 0.6759 | 0.7250 | 0.7706 | 0.7835 |
| UWavZ | **0.8492** | 0.7674 | 0.6955 | 0.9422 | 0.7725 | 0.7548 | 0.7334 | 0.6635 | 0.7751 | 0.7912 | 0.7958 | 0.7064 | 0.7514 | 0.7290 | 0.7732 | 0.7918 |
| Wafer | 0.9778 | 0.9961 | 0.9989 | **1.0000** | **1.0000** | 0.9999 | 0.9966 | 0.9954 | 0.9999 | 0.9989 | 0.9986 | 0.9973 | 0.9989 | 0.9970 | 0.9986 | **1.0000** |
| Wine | 0.5185 | 0.8562 | 0.8926 | 0.7963 | 0.8858 | **0.9302** | 0.8623 | 0.8710 | 0.8920 | 0.8981 | 0.9142 | 0.7000 | 0.8562 | 0.8890 | 0.8864 | 0.6333 |
| WordSyn | 0.6803 | 0.7782 | 0.6584 | 0.5705 | 0.6230 | 0.7127 | 0.6479 | 0.5916 | 0.6932 | **0.7937** | 0.7644 | 0.5440 | 0.6133 | 0.5800 | 0.7518 | 0.6605 |
| Worms | 0.5844 | 0.6931 | 0.7255 | 0.7403 | 0.7333 | 0.7784 | 0.6121 | 0.6866 | 0.7165 | 0.7684 | 0.7203 | 0.7251 | 0.7589 | 0.6690 | **0.7797** | 0.6522 |
| WormsTC | 0.6364 | 0.7693 | 0.8078 | **0.8312** | 0.7853 | 0.8004 | 0.6935 | 0.7853 | 0.7896 | 0.7861 | 0.7896 | 0.7922 | 0.7680 | 0.7290 | 0.8035 | 0.7946 |
| Yoga | 0.8200 | 0.8874 | 0.9102 | 0.8177 | 0.8800 | 0.8924 | 0.8658 | 0.8372 | **0.9124** | 0.8726 | 0.9138 | 0.8038 | 0.8772 | 0.8450 | 0.9123 | 0.9007 |
| AVG Rank | 13.3839 | 8.0092 | 8.8661 | 9.0625 | 7.1161 | 7.7589 | 10.4107 | 10.0446 | 4.4375 | 4.6239 | 5.0183 | 11.6071 | 7.3482 | 7.8750 | 5.3395 | 6.5982 |
| MPSE | 0.0746 | 0.0438 | 0.0390 | 0.0425 | 0.0349 | 0.0375 | 0.0481 | 0.0462 | 0.0293 | 0.0321 | 0.0355 | 0.0510 | 0.0398 | 0.0372 | 0.0320 | 0.0380 |

Among all methods, HIVE-COTE achieved the best performance. Comparing the two distance-based methods, the accuracy of PF is much improved compared to DTW. The difference between the two is that PF uses multiple distance functions. Among the feature-based methods, STC achieved the best performance. What makes STC special is that it uses some innovative methods to find more effective shapelet features. In the deep learning methods, the best performance is achieved by InceptionTime. InceptionTime combines several DNNs with the same structure but different parameters, and it can be considered that multiple DNNs learn different features, thus InceptionTime improved the performance compared to other deep learning methods.

Those achieving excellent performance methods either combine multiple distance functions and features, or find and select more effective features, or use multiple DNNs to learn various features from limited data. The common point of these methods is to use more effective features (or more features to cover the most effective features) to improve performance in the case of limited data. Similarly, the proposed method TSC-FF purposefully selected and used the fusion features of multiple features learned from different aspects, so it achieved promising results.

In the Wilcoxon Sign Rank Test (Table 3), the p-values of TSC-FF with 1NN_DTW, TSF, RISE and Catch22 were smaller than 0.05, while with other methods were larger than 0.05, indicating that TSC-FF had almost equally good results as those achieving excellent performance methods. The reason why TSC-FF did not achieve the best overall result is that TSC-FF is a method based on deep learning. TSC-FF does not suffer from high computational complexity, due to employing multiple different classifiers, compared with the ensemble method HIVE-COTE, but the sub models used by TSC-FF are more affected by overfitting due to the training set being too small, considering its difference in structure compared to InceptionTime, FCN and ResNet. Some data sets in UCR had a small training set and a large number of categories, such that the sub-models were easily over-fitted, resulting in insufficient extracted features and affecting the classification performance.

We used the DNN to learn the characteristics of different classes. Whether there are enough records for each class to learn is one of the problems that must be considered. Table 4 shows the performance of all methods with different amounts of records in each category. The methods based on deep learning achieved the best performance. In particular, InceptionTime, which used multiple DNNs, achieved the highest average accuracy in most cases with different data volumes in each category. This shows that DNNs can learn more effective features than artificial features and replace the work of constructing features by human beings.

In our proposed framework, TS data was converted into Area Graphs to extract visualization features. However, some existing methods convert TS data into image data through other different methods to extract features, so we also carried out a comparison with these methods. We chose two such methods: The first method uses Tiled Convolutional Neural Networks (tiled CNNs) on Gramian Angular Summation/Difference Fields (GASF/GADF) and MTF to extract features for classification, abbreviated as GASF-GADF-MTF. The second method uses a Support Vector Machine (SVM) on Texture Features extracted from RP for classification, abbreviated as TFRP. Table 5 shows the comparison results. Through the comparison, we can see that the proposed framework had better performance than the selected comparison methods. The reason for this may be that the visualization method in this paper is simple, avoiding the loss of some information in the conversion process. In addition, the combination of different aspects of features (i.e., sequence features and visualization features) makes up for the deficiencies of single-category features.

**Table 3.** Wilcoxson signed-rank test comparision of each model; the significance level (alpha) of 0.05 was selected for all statistical tests, bold values indicate smaller than 0.05.

| | IcT | FCN | RN | C2 | RK | CHI | HCT | RISE | TSF | WS | STC | ST | BOSS | PF | DTW |
|---|---|---|---|---|---|---|---|---|---|---|---|---|---|---|---|
| FCN | $\mathbf{1.5 \times 10^{-2}}$ | | | | | | | | | | | | | | |
| RN | $2.3 \times 10^{-1}$ | $2.0 \times 10^{-1}$ | | | | | | | | | | | | | |
| C2 | $\mathbf{6.6 \times 10^{-6}}$ | $6.2 \times 10^{-2}$ | $\mathbf{1.4 \times 10^{-3}}$ | | | | | | | | | | | | |
| RK | $7.8 \times 10^{-1}$ | $\mathbf{2.9 \times 10^{-2}}$ | $4.1 \times 10^{-1}$ | $\mathbf{2.6 \times 10^{-5}}$ | | | | | | | | | | | |
| CHI | $6.3 \times 10^{-1}$ | $\mathbf{6.7 \times 10^{-3}}$ | $1.1 \times 10^{-1}$ | $\mathbf{3.2 \times 10^{-6}}$ | $4.5 \times 10^{-1}$ | | | | | | | | | | |
| HCT | $4.5 \times 10^{-1}$ | $\mathbf{1.7 \times 10^{-3}}$ | $6.0 \times 10^{-2}$ | $\mathbf{1.4 \times 10^{-7}}$ | $2.8 \times 10^{-1}$ | $8.1 \times 10^{-1}$ | | | | | | | | | |
| RISE | $\mathbf{4.8 \times 10^{-4}}$ | $3.9 \times 10^{-1}$ | $\mathbf{2.6 \times 10^{-2}}$ | $2.8 \times 10^{-1}$ | $\mathbf{1.2 \times 10^{-3}}$ | $\mathbf{2.3 \times 10^{-4}}$ | $\mathbf{1.6 \times 10^{-5}}$ | | | | | | | | |
| TSF | $\mathbf{1.8 \times 10^{-3}}$ | $4.6 \times 10^{-1}$ | $\mathbf{4.6 \times 10^{-2}}$ | $2.8 \times 10^{-1}$ | $\mathbf{3.9 \times 10^{-3}}$ | $\mathbf{6.0 \times 10^{-4}}$ | $\mathbf{7.6 \times 10^{-5}}$ | $9.1 \times 10^{-1}$ | | | | | | | |
| WS | $1.8 \times 10^{-1}$ | $1.8 \times 10^{-1}$ | $8.9 \times 10^{-1}$ | $\mathbf{6.0 \times 10^{-4}}$ | $3.1 \times 10^{-1}$ | $7.4 \times 10^{-2}$ | $\mathbf{3.3 \times 10^{-2}}$ | $\mathbf{1.7 \times 10^{-2}}$ | $\mathbf{4.3 \times 10^{-2}}$ | | | | | | |
| STC | $3.6 \times 10^{-1}$ | $8.6 \times 10^{-2}$ | $8.2 \times 10^{-1}$ | $\mathbf{8.2 \times 10^{-5}}$ | $5.7 \times 10^{-1}$ | $1.7 \times 10^{-1}$ | $7.5 \times 10^{-2}$ | $\mathbf{3.8 \times 10^{-3}}$ | $\mathbf{1.3 \times 10^{-2}}$ | $6.8 \times 10^{-1}$ | | | | | |
| ST | $\mathbf{4.2 \times 10^{-2}}$ | $5.6 \times 10^{-1}$ | $3.8 \times 10^{-1}$ | $\mathbf{7.9 \times 10^{-3}}$ | $7.5 \times 10^{-2}$ | $\mathbf{1.6 \times 10^{-2}}$ | $\mathbf{4.8 \times 10^{-3}}$ | $1.0 \times 10^{-1}$ | $1.8 \times 10^{-1}$ | $4.1 \times 10^{-1}$ | $2.1 \times 10^{-1}$ | | | | |
| BOSS | $2.0 \times 10^{-1}$ | $2.7 \times 10^{-1}$ | $8.6 \times 10^{-1}$ | $\mathbf{2.3 \times 10^{-3}}$ | $2.9 \times 10^{-1}$ | $8.5 \times 10^{-2}$ | $\mathbf{3.2 \times 10^{-2}}$ | $\mathbf{4.0 \times 10^{-2}}$ | $5.9 \times 10^{-2}$ | $8.7 \times 10^{-1}$ | $5.9 \times 10^{-1}$ | $5.8 \times 10^{-1}$ | | | |
| PF | $8.7 \times 10^{-2}$ | $5.3 \times 10^{-1}$ | $5.9 \times 10^{-1}$ | $\mathbf{1.1 \times 10^{-2}}$ | $1.5 \times 10^{-1}$ | $\mathbf{3.7 \times 10^{-2}}$ | $\mathbf{1.4 \times 10^{-2}}$ | $1.1 \times 10^{-1}$ | $1.6 \times 10^{-1}$ | $6.1 \times 10^{-1}$ | $3.5 \times 10^{-1}$ | $8.8 \times 10^{-1}$ | $6.9 \times 10^{-1}$ | | |
| DTW | $\mathbf{1.4 \times 10^{-10}}$ | $\mathbf{2.2 \times 10^{-5}}$ | $\mathbf{1.3 \times 10^{-7}}$ | $\mathbf{1.7 \times 10^{-3}}$ | $\mathbf{5.1 \times 10^{-10}}$ | $\mathbf{7.0 \times 10^{-11}}$ | $\mathbf{5.4 \times 10^{-13}}$ | $\mathbf{6.8 \times 10^{-5}}$ | $\mathbf{1.3 \times 10^{-4}}$ | $\mathbf{5.0 \times 10^{-9}}$ | $\mathbf{3.0 \times 10^{-10}}$ | $\mathbf{9.8 \times 10^{-8}}$ | $\mathbf{9.1 \times 10^{-8}}$ | $\mathbf{7.6 \times 10^{-7}}$ | |
| TSC-FF | $4.6 \times 10^{-1}$ | $1.4 \times 10^{-1}$ | $7.2 \times 10^{-1}$ | $\mathbf{7.8 \times 10^{-4}}$ | $6.3 \times 10^{-1}$ | $2.8 \times 10^{-1}$ | $1.4 \times 10^{-1}$ | $\mathbf{1.8 \times 10^{-2}}$ | $\mathbf{2.6 \times 10^{-2}}$ | $7.3 \times 10^{-1}$ | $9.3 \times 10^{-1}$ | $3.0 \times 10^{-1}$ | $6.0 \times 10^{-1}$ | $3.7 \times 10^{-1}$ | $\mathbf{2.7 \times 10^{-8}}$ |

**Table 4.** Average accuracy of all seven methods with different amounts of records in each category. Bold values denote the model with the best performance.

| Records in Each Category | (0,20] | (20,50] | (50,100] | (100,200] | (200,500] | (500,1000] | (1000,+∞) |
|---|---|---|---|---|---|---|---|
| Distance-based methods | | | | | | | |
| DTW | 0.7208 | 0.7186 | 0.7433 | 0.6497 | 0.8156 | 0.7401 | 0.6512 |
| PF | 0.8593 | 0.8311 | 0.8745 | 0.7472 | 0.9003 | 0.2755 | 0.8439 |
| Feature-based methods | | | | | | | |
| BOSS | 0.8703 | 0.8148 | 0.8463 | 0.7543 | 0.8917 | 0.8174 | 0.8754 |
| ST | 0.8574 | 0.8377 | 0.8417 | 0.7585 | 0.8660 | 0.7634 | 0.8419 |
| STC | 0.8623 | 0.8497 | 0.8676 | 0.8110 | 0.9059 | 0.8337 | 0.9119 |
| WS | 0.8721 | 0.8374 | 0.8350 | 0.7734 | 0.9037 | 0.8217 | 0.9126 |
| TSF | 0.7704 | 0.8077 | 0.8689 | 0.7436 | 0.8927 | 0.8057 | 0.7999 |
| RISE | 0.7998 | 0.7853 | 0.8577 | 0.7483 | 0.8315 | 0.8125 | 0.8938 |
| Ensemble methods | | | | | | | |
| HCT | 0.8977 | 0.8687 | **0.8903** | 0.8232 | 0.9067 | 0.8265 | 0.9173 |
| CHI | 0.9147 | 0.8543 | 0.8844 | 0.7862 | 0.9021 | 0.9601 | 0.9106 |
| RK | 0.9001 | 0.8484 | 0.8679 | 0.8074 | 0.9085 | 0.1955 | 0.9195 |
| C2 | 0.7721 | 0.7638 | 0.8347 | 0.7375 | 0.8532 | 0.7919 | 0.8838 |
| Deep learning methods | | | | | | | |
| RN | 0.8060 | **0.8728** | 0.8081 | 0.7957 | 0.9056 | 0.8477 | 0.9109 |
| FCN | 0.7936 | 0.8358 | 0.7660 | 0.7710 | 0.8741 | 0.8260 | 0.8373 |
| Ict | **0.9009** | 0.8597 | 0.8439 | **0.8137** | **0.9127** | **0.9221** | **0.9301** |
| TSC-FF | 0.8333 | 0.8640 | 0.8324 | 0.8027 | 0.8787 | 0.8096 | 0.9138 |

**Table 5.** Accuracy comparison of visual feature-based methods on 20 data sets in UCR. Bold values denote model with the best performance.

| | GASF-GADF-MTF | TFRP | TSC-FF |
|---|---|---|---|
| Adiac | 0.6270 | **0.7954** | 0.7875 |
| Beef | 0.7670 | 0.6333 | **0.8444** |
| CBF | 0.9910 | N/A | **1.0000** |
| Coffee | **1.0000** | 0.9643 | **1.0000** |
| ECGFiveDays | 0.9100 | 0.8955 | **0.9991** |
| FaceAll | 0.7630 | 0.7101 | **0.8705** |
| FaceFour | **0.9320** | 0.7841 | 0.9091 |
| 50words | 0.6990 | 0.5626 | **0.8041** |
| fish | 0.8860 | 0.8800 | **0.9886** |
| Gun Point | 0.9200 | 0.9800 | **1.0000** |
| Lighting2 | 0.8860 | 0.8525 | **1.0000** |
| Lighting7 | 0.7400 | 0.6849 | **0.8115** |
| OliveOil | 0.8000 | 0.8667 | **0.8667** |
| OSULeaf | 0.6420 | 0.9298 | **0.9695** |
| SwedishLeaf | 0.9350 | **0.9504** | 0.8779 |
| SyntheticControl | 0.9930 | N/A | **1.0000** |
| Trace | **1.0000** | N/A | **1.0000** |
| Two Patterns | 0.9090 | N/A | **1.0000** |
| wafer | **1.0000** | 0.9998 | **1.0000** |
| yoga | 0.8040 | 0.8587 | **0.9007** |
| Average Rank | 2.0500 | 2.3125 | 1.2500 |
| MPSE | $2.6500 \times 10^{-2}$ | $3.1100 \times 10^{-2}$ | $9.8900 \times 10^{-3}$ |

*4.3. Sub-Model Evaluation*

Figure 6 depicts the accuracy and loss curves of two sub-models trained on the data set StarLightCurves. From the figure, we can see that, although the accuracy rate and loss fluctuated, the overall accuracy rate gradually increased and the loss gradually decreased, finally reaching a stable state, and that the results on the training set were slightly better than the results of the verification set, which shows that the model has good generalization ability.

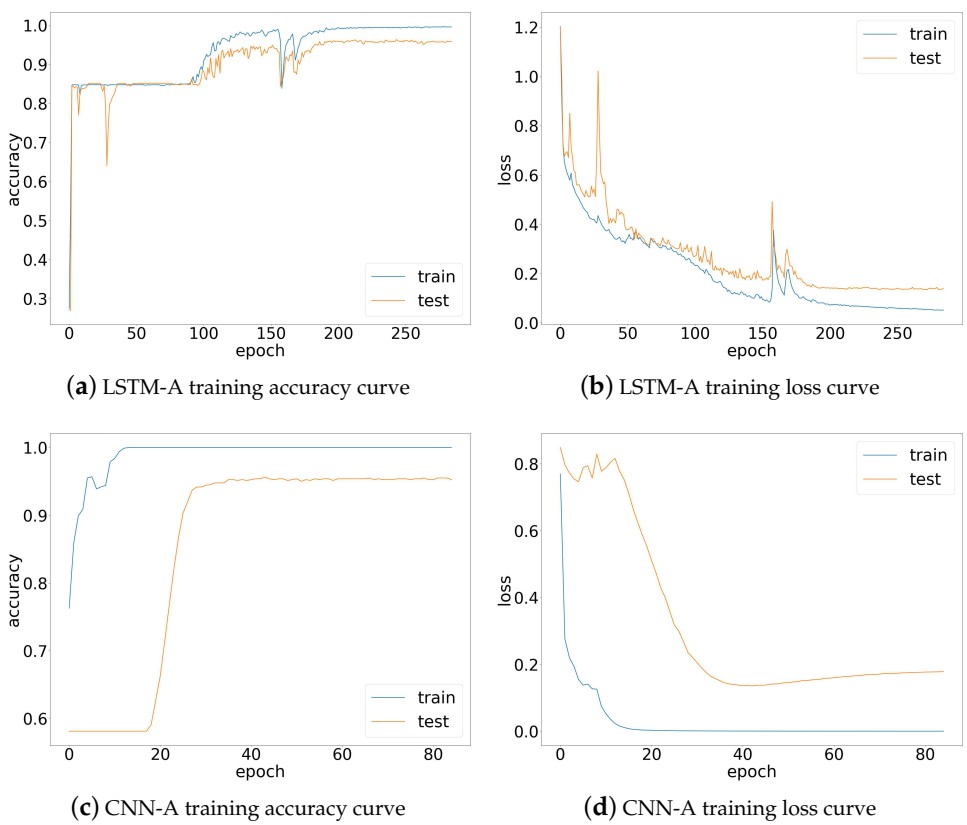

(**a**) LSTM-A training accuracy curve

(**b**) LSTM-A training loss curve

(**c**) CNN-A training accuracy curve

(**d**) CNN-A training loss curve

**Figure 6.** Accuracy and loss curves of training two sub-models on the data set StarLightCurves.

Figure 4 shows an example of transformed Area Graphs and features from the 'BeetleFly' data set. In the data set BeetleFly, the records are divided into two categories. We randomly selected one record from the validation set of each category and show the corresponding Area Graphs (Figure 4a,b), sequence features (Figure 4c,d), and visualization features (Figure 4e,f) extracted by LSTM-A and CNN-A, respectively. Using the area in the red box in Figure 4a,b for comparison, the numbers in this area in the original record were quite different, which may be the main trend used to distinguish record categories. Correspondingly, in the learned sequence features, the corresponding positions were the wave crest and wave trough; that is, they were quite different. In the visualization features, we can see that most of the features of the corresponding positions were similar, as the numbers around the red box in the original record are similar, and only one part of the visual features corresponding to the value of the red box position is different, which proves that CNN-A has learned the features of similar parts, as well as the features of the different parts, of the record.

We also compared the accuracy of the pre-trained model with that of the integrated model. Table 6 shows the accuracy comparison of the two sub-models and the integrated model on 112 data sets. The bold values in TSC-FF denote model wins or ties with sub-models, while those in LSTM-A and CNN-A denote model wins or ties with LSTM and CNN, respectively. Among them, the integrated model TSC-FF achieved a higher accuracy on 72 data sets than the two sub-models. This shows that the combination of the two features generally improved the classification performance. By comparing

the accuracy of the sub-models with and without the attention mechanism, we can see that LSTM-A achieved higher accuracy on 95 data sets than without the attention mechanism, and the same for CNN-A was 103. This indicates that, for the two sub-models, when using the attention mechanism, the model can learn more effective features. In addition, the difference in performance between CNN-A and CNN also illustrates the effectiveness of the CT-Attention method proposed in this paper.

**Table 6.** Accuracy comprison of TSC-FF with sub-models on 112 data sets.

|  | TSC-FF | LSTM-A | LSTM | CNN-A | CNN |
|---|---|---|---|---|---|
| ACSF1 | 0.8552 | **0.8278** | 0.7800 | **0.8660** | 0.8400 |
| Adiac | **0.7875** | **0.7684** | 0.6199 | **0.7399** | 0.6513 |
| ArrowHead | 0.8025 | **0.8586** | 0.7580 | **0.8185** | 0.6822 |
| Beef | 0.8444 | **0.8704** | 0.8222 | 0.7852 | 0.7963 |
| BeetleFly | **0.8889** | **0.8333** | 0.6111 | **0.8121** | 0.5023 |
| BirdChicken | 0.9222 | **0.9444** | 0.8423 | 0.7342 | 0.7932 |
| BME | 0.8921 | **0.8400** | 0.8333 | **0.9200** | 0.9133 |
| Car | 0.9500 | **0.9667** | 0.8852 | **0.8667** | 0.8222 |
| CBF | **1.0000** | **0.9989** | 0.8593 | **1.0000** | 0.6716 |
| Chinatown | **0.9861** | 0.9478 | 0.9536 | **0.9652** | 0.9217 |
| ChlCon | **0.7246** | **0.6892** | 0.6399 | **0.7127** | 0.6306 |
| CinCECGtorso | 0.9511 | **0.8719** | 0.8300 | **0.9778** | 0.8911 |
| Coffee | **1.0000** | **1.0000** | 1.0000 | **1.0000** | 1.0000 |
| Computers | **0.7200** | **0.5600** | 0.5400 | **0.7200** | 0.6022 |
| CricketX | **0.8077** | **0.7923** | 0.7356 | **0.7694** | 0.7279 |
| CricketY | **0.8000** | **0.7556** | 0.6400 | **0.7410** | 0.5761 |
| CricketZ | **0.8282** | **0.7926** | 0.5499 | **0.7667** | 0.5627 |
| Crop | 0.7622 | **0.7636** | 0.7589 | **0.7712** | 0.7640 |
| DiaSizeRed | **0.9471** | **0.8800** | 0.5673 | **0.9239** | 0.4982 |
| DislPhaOutCor | **0.7661** | 0.7460 | 0.7621 | **0.7661** | 0.5968 |
| DisPhaOutAgeGro | **0.7120** | **0.6960** | 0.4000 | **0.7040** | 0.4240 |
| DisPhaTW | 0.6875 | **0.6960** | 0.4960 | **0.6480** | 0.4800 |
| Earthquakes | **0.8160** | **0.8080** | 0.7200 | 0.7360 | 0.7440 |
| ECG200 | **0.8916** | **0.8724** | 0.8245 | **0.8869** | 0.8046 |
| ECG5000 | 0.9481 | **0.9427** | 0.7230 | **0.9486** | 0.9442 |
| ECGFiveDays | 0.9991 | 0.7356 | 0.8052 | **0.9998** | 0.7935 |
| ElectricDevices | 0.7413 | **0.7420** | 0.7381 | **0.6938** | 0.6757 |
| EOGHorSig | **0.8358** | **0.7818** | 0.7569 | **0.8011** | 0.7707 |
| EOGVerSig | **0.7722** | **0.7541** | 0.7348 | **0.7569** | 0.7486 |
| EthLevel | **0.8442** | **0.8085** | 0.7864 | **0.8182** | 0.7485 |
| FaceAll | 0.8705 | **0.8567** | 0.7344 | **0.8824** | 0.7879 |
| FaceFour | **0.9091** | **0.8608** | 0.2785 | **0.8861** | 0.2785 |
| FacesUCR | 0.9593 | **0.9786** | 0.8618 | **0.8390** | 0.8119 |
| FiftyWords | **0.8041** | **0.7822** | 0.4318 | **0.7922** | 0.4320 |
| Fish | **0.9886** | **0.9471** | 0.8019 | **0.9134** | 0.8338 |
| FordA | **1.0000** | **1.0000** | 1.0000 | **1.0000** | 1.0000 |
| FordB | **1.0000** | **1.0000** | 1.0000 | **1.0000** | 1.0000 |
| FreRegTra | **0.9989** | 0.9940 | 0.9961 | **0.9979** | 0.9965 |
| FreSmaTra | **0.9668** | **0.9660** | 0.9547 | **0.9642** | 0.9323 |
| GunPoint | **1.0000** | **0.9556** | 0.6889 | **0.9852** | 0.5333 |
| GPAgSpa | **1.0000** | **1.0000** | 1.0000 | **1.0000** | 1.0000 |
| GPMalVerFem | **1.0000** | **1.0000** | 1.0000 | **1.0000** | 0.9968 |
| GPOldVerYou | **0.9264** | **0.9244** | 0.9199 | **0.9011** | 0.8507 |
| Ham | **0.7128** | **0.6489** | 0.6170 | **0.7021** | 0.4787 |
| HandOutlines | **0.9069** | **0.8679** | 0.3634 | **0.8739** | 0.6366 |
| Haptics | **0.5213** | 0.3863 | 0.3971 | **0.4621** | 0.2986 |
| Herring | **0.6491** | **0.6316** | 0.6140 | **0.5088** | 0.3860 |
| HouTwe | 0.9613 | **0.9412** | 0.8824 | **0.9664** | 0.9496 |
| InlineSkate | 0.4616 | 0.3747 | 0.4404 | **0.4694** | 0.4040 |
| InsEPGRegTra | **1.0000** | **1.0000** | 1.0000 | **1.0000** | 1.0000 |
| InsEPGSmaTra | 0.5150 | **0.4739** | 0.4626 | **0.5238** | 0.4610 |
| InsWinSou | **0.6269** | **0.5752** | 0.5196 | **0.6066** | 0.3889 |

**Table 6.** *Cont.*

|  | TSC-FF | LSTM-A | LSTM | CNN-A | CNN |
|---|---|---|---|---|---|
| ItaPowDem | **0.9633** | 0.9514 | 0.9568 | **0.9536** | 0.4989 |
| LarKitApp | **0.8723** | 0.8249 | 0.8398 | **0.8567** | 0.8205 |
| Lightning2 | 0.8155 | **0.8126** | 0.7955 | **0.8175** | 0.7246 |
| Lightning7 | **0.8115** | 0.8077 | 0.7231 | **0.7923** | 0.7769 |
| Mallat | **0.9542** | 0.9095 | 0.8313 | **0.8948** | 0.8564 |
| Meat | 0.9333 | 0.9148 | 0.9148 | **0.9407** | 0.8519 |
| MedicalImages | **0.7578** | 0.7442 | 0.7456 | **0.6988** | 0.5146 |
| MidPhaOutCor | 0.7663 | **0.7778** | 0.5556 | **0.7203** | 0.5556 |
| MidPhaOutAgeGro | **0.7000** | 0.5942 | 0.5000 | 0.5217 | 0.5725 |
| MidPhalanxTW | **0.5745** | 0.5652 | 0.4725 | **0.5072** | 0.4812 |
| MixShaRegTra | 0.9690 | **0.9711** | 0.9691 | **0.8613** | 0.8566 |
| MixShaSmaTra | **0.9386** | 0.9185 | 0.8818 | **0.9209** | 0.9177 |
| MoteStrain | **0.8943** | 0.8757 | 0.3357 | **0.8535** | 0.5444 |
| NonInvFatECGTho1 | 0.8818 | **0.9143** | 0.7766 | **0.8415** | 0.7538 |
| NonInvFatECGTho2 | **0.9424** | 0.9394 | 0.8305 | **0.9100** | 0.8236 |
| OliveOil | 0.8667 | **0.8852** | 0.8704 | **0.8407** | 0.7704 |
| OSULeaf | 0.9699 | 0.8747 | 0.8346 | **0.9712** | 0.8502 |
| PhaOutlCor | 0.8096 | **0.8187** | 0.6088 | **0.7293** | 0.6088 |
| Phoneme | **0.3058** | 0.2914 | 0.2492 | **0.2728** | 0.2400 |
| PigAirPre | 0.5370 | **0.5529** | 0.4183 | **0.5673** | 0.5106 |
| PigArtPre | 0.9719 | 0.9712 | 0.9279 | **0.9760** | 0.9663 |
| PigCVP | 0.5253 | **0.4933** | 0.4365 | **0.5444** | 0.5198 |
| Plane | **1.0000** | 0.9574 | 0.8489 | **0.9894** | 0.7957 |
| PowerCons | **0.9601** | 0.9111 | 0.9056 | 0.8833 | 0.8944 |
| ProPhaOutCor | **0.8538** | 0.7280 | 0.6858 | **0.7854** | 0.6858 |
| ProPhaOutAgeGro | **0.8761** | 0.7891 | 0.8326 | **0.8370** | 0.7707 |
| ProPhalanxTW | **0.8043** | 0.7533 | 0.7489 | **0.7989** | 0.7315 |
| RefrigeDevices | **0.5697** | 0.4362 | 0.4807 | **0.5460** | 0.3412 |
| Rock | **0.6133** | 0.4213 | 0.3756 | **0.4265** | 0.3961 |
| ScreenType | **0.5392** | 0.4620 | 0.4214 | **0.4887** | 0.3917 |
| SHGenCh2 | 0.9521 | **0.8133** | 0.8000 | **0.9651** | 0.9517 |
| SHMovCh2 | 0.5916 | **0.4244** | 0.3956 | **0.6044** | 0.4156 |
| SHSubCh2 | 0.6121 | **0.5556** | 0.5289 | **0.6167** | 0.5644 |
| ShapeletSim | 0.8062 | **0.8123** | 0.7938 | **0.7938** | 0.7338 |
| ShapesAll | **0.8148** | 0.8148 | 0.7407 | **0.7648** | 0.7870 |
| SmaKitApp | 0.8146 | **0.8020** | 0.6024 | **0.8167** | 0.6205 |
| SmoSub | 0.9833 | **0.9933** | 0.9867 | **0.5455** | 0.5195 |
| SonAIBORobSur1 | 0.7630 | **0.7759** | 0.7759 | **0.7259** | 0.7241 |
| SonAIBORobSur2 | **0.9016** | 0.8734 | 0.8046 | **0.9016** | 0.8254 |
| StarlightCurves | **0.9799** | 0.9541 | 0.9611 | **0.9493** | 0.8562 |
| Strawberry | **0.9709** | 0.9577 | 0.9347 | **0.9259** | 0.8577 |
| SwedishLeaf | 0.8779 | **0.8986** | 0.8772 | **0.8505** | 0.8270 |
| Symbols | 0.8000 | **0.8078** | 0.7888 | **0.7598** | 0.6536 |
| SyntheticControl | **1.0000** | 0.9852 | 0.9926 | **1.0000** | 0.3296 |
| ToeSeg1 | **0.9793** | 0.9268 | 0.8951 | **0.9724** | 0.9463 |
| ToeSeg2 | 0.9318 | 0.8863 | 0.9162 | **0.9447** | 0.8966 |
| Trace | **1.0000** | 0.9889 | 0.9889 | **1.0000** | 0.7000 |
| TwoLeadECG | 0.9288 | 0.7893 | 0.5034 | **0.9717** | 0.5034 |
| TwoPatterns | **1.0000** | 1.0000 | 0.9994 | **0.9989** | 0.2478 |
| UMD | 0.8885 | **0.9583** | 0.9475 | **0.8667** | 0.6478 |
| UWaveGesLibAll | **0.9705** | 0.9507 | 0.7415 | 0.9656 | 0.9693 |
| UWaveGesLibX | 0.8460 | **0.8177** | 0.7813 | **0.7986** | 0.7821 |
| UWaveGesLibY | **0.7835** | 0.7817 | 0.6665 | 0.7009 | 0.7287 |
| UWaveGesLibZ | **0.7918** | 0.7581 | 0.7056 | **0.7173** | 0.6440 |
| Wafer | **1.0000** | 1.0000 | 1.0000 | **1.0000** | 1.0000 |
| Wine | **0.6333** | 0.5000 | 0.6000 | **0.6333** | 0.6000 |
| WordSynonyms | 0.6605 | **0.6348** | 0.6254 | **0.7145** | 0.6164 |
| Worms | **0.6522** | 0.5217 | 0.5942 | 0.4493 | 0.5217 |
| WormsTwoClass | **0.7946** | 0.7522 | 0.7362 | **0.7377** | 0.7348 |
| Yoga | **0.9007** | 0.8933 | 0.7948 | **0.8930** | 0.5359 |

Finally, we explored the influence of the length of TS records on the feature learning of sub-models. From Table 7, we can see that, when the record length was small (<80), the average test accuracy obtained by LSTM-A was higher than that of CNN-A and, as the record length increased, CNN-A performed better than LSTM-A. This shows that the sequence features learned by LSTM-A on short TS records were more effective than the visual features learned by CNN-A. This is because, as the record length increases, the LSTM-A may "forget" some features, while using CNN-A to learn visualization features is not affected by this. In addition, when we visualize long data, some small fluctuations will not be clearly displayed on the picture. Therefore, CNN-A can learn some more distinguishable features, such as the main trends of the data.

**Table 7.** Average test accuracy grouped by data set length.

| Length | <80 | 81–250 | 251–450 | 451–700 | 701–1000 | >1000 |
|--------|--------|--------|---------|---------|----------|--------|
| LSTM-A | 0.7915 | 0.8840 | 0.8413 | 0.8658 | 0.7060 | 0.7143 |
| CNN-A  | 0.7577 | 0.8957 | 0.8496 | 0.8186 | 0.7205 | 0.7472 |

## 5. Conclusions and Future Work

In this paper, we used a DNN to learn features from different aspects to enhance the feature space. Specifically, we used LSTM-A to learn sequence features from raw TS data and CNN-A to learn visual features from Area Graphs converted from TS data. Then, we classified TS data based on the fused features. Through various forms of comparison and analysis, we found that the well-trained LSTM-A and CNN-A learned features that could effectively distinguish the TS data and promote each other. With the proposed framework, we have proved that, in the task of TSC, using deep learning methods can achieve similar performance as the complex ensemble methods, and the features extracted by deep learning are more effective and general than artificially constructed features.

We proposed the use of CNN-based methods to extract visualization features, which is feasible for univariate TS data; however, on multivariate TS data, how to perform visualization to extract visual features and how to handle the correlation between multi-dimensional data provides difficulties which need to be solved for the framework proposed in this paper to expand on multivariate TS data.

**Supplementary Materials:** Our source code has been uploaded to https://github.com/wangbaoquan520/TSC-FF.

**Author Contributions:** Conceptualization, B.W.; Data curation, B.M.; Formal analysis, B.W.; Funding acquisition, T.J., X.Z.; Investigation, B.W.; Methodology, B.W.; Project administration, H.J.; Resources, B.W.; Software, B.W.; Supervision, F.Z.; Validation, B.W.; Visualization, F.Z.; Writing—original draft, B.W.; Writing—review & editing, B.M., Y.W. and B.W. All authors have read and agreed to the published version of the manuscript.

**Funding:** This research was funded by The Xinjiang Science and Technology Major Project, Grant No.2016A03007-2, The Youth Innovation Promotion Association of Chinese Academy of Sciences, Grant No.Y9290802, The West Light Foundation of The Chinese Academy of Sciences, Grant No.2019-XBQNXZ-A-004, The Tianshan Excellent Young Scholars of Xinjiang, Grant No.2018Q032.

**Conflicts of Interest:** The authors declare no conflict of interest.

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
