# Peer review of "Time-Series Classification Based on Fusion Features of Sequence and Visualization"

_applsci, doi:10.3390/app10124124_

Round 1

Reviewer 1 Report

I decide that the paper requires a major revision for the following reason.  

The contribution of this paper should be explained more specifically. For example, the specialty (or superiority) of the transformation method of the raw data to image data in this paper.

There still exist a lot of grammatical errors and typos. Authors should carefully check the full text again. I suggest the authors get a commercial English editing service to improve the readability.  

No Figure 7 can be found in this paper.

It seems necessary to unify the decimal point in the tables.

Reviewer 2 Report

Authors have answered and solved most of the questions.

However, there is still an unsolved question. In the experimental section, authors have used results published in other works without reproducing them themselves. This has reduced the amount of datasets from 128 to 85. Being this a work with a high experimental load, it is necessary to reproduce the results used and to increase the amount of processed datasets. Especially if the state of the art is the 128 datasets mentioned above.

Checking for some typing errors:

reocrd

HIVE-COTE,FCN

Reviewer 3 Report

Comparative evaluation shows that the method does not beat the state-of-the-art and at the same time is computationally very expensive.

Round 2

Reviewer 1 Report

The manuscript has been revised as commented. 

Author Response

Thank you for your comments.

Reviewer 2 Report

The authors have answered and solved the questions raised.

Author Response

Thank you for your comments.

Reviewer 3 Report

The author did not consider in their discussion on the SOTA all relevant methods which I mentioned in my original review. 
